

# Enhancing geophysical flow machine learning performance via scale separation

**Davide Faranda**[1,2,3], **Mathieu Vrac**[1], **Pascal Yiou**[1], **Flavio Maria Emanuele Pons**[1], **Adnane Hamid**[1], **Giulia Carella**[1], **Cedric Ngoungue Langue**[1], **Soulivanh Thao**[1], and **Valerie Gautard**[4]

[1]Laboratoire des Sciences du Climat et de l'Environnement, CE Saclay l'Orme des Merisiers, UMR 8212
CEA-CNRS-UVSQ, Université Paris-Saclay & IPSL, 91191 Gif-sur-Yvette, France
[2]London Mathematical Laboratory, 8 Margravine Gardens, London, W68RH, UK
[3]LMD/IPSL, Ecole Normale Superieure, PSL research University, Paris, France
[4]DRF/IRFU/DEDIP//LILAS Departement d'Electronique des Detecteurs et d'Informatique pour la Physique,
CE Saclay l'Orme des Merisiers, 91191 Gif-sur-Yvette, France.

**Correspondence:** Davide Faranda (davide.faranda@lsce.ipsl)

**Abstract.** Recent advances in statistical and machine learning have opened the possibility of forecasting the behaviour of chaotic systems using recurrent neural networks. In this article we investigate the applicability of such a framework to geophysical flows, known to involve multiple scales in length, time and energy and to feature intermittency. We show that both multiscale dynamics and intermittency introduce severe limitations on the applicability of recurrent neural networks, both for short-term forecasts as well as for the reconstruction of the underlying attractor. We suggest that possible strategies to overcome such limitations should be based on separating the smooth large-scale dynamics from the intermittent/small-scale features. We test these ideas on global sea-level pressure data for the past 40 years, a proxy of the atmospheric circulation dynamics. Better short- and long-term forecasts of sea-level pressure data can be obtained with an optimal choice of spatial coarse graining and time filtering.

## 1 Introduction

The advent of high-performance computing has paved the way for advanced analyses of high-dimensional datasets (Jordan and Mitchell, 2015; LeCun et al., 2015). Those successes have naturally raised the question of whether it is possible to learn the behaviour of a dynamical system without resolving or even without knowing the underlying evolution equations. Such an interest is motivated on the one hand by the fact that many complex systems still miss a universally accepted state equation – e.g. brain dynamics (Bassett and Sporns, 2017) and macro-economical and financial systems (Quinlan et al., 2019) – and, on the other, by the need to reduce the complexity of the dynamical evolution for the systems for which the underlying equations are known – e.g. on geophysical and turbulent flows (Wang et al., 2017). Evolution equations are difficult to solve for large systems such as the geophysical flows, so that approximations and parameterizations are needed for meteorological and climatological applications (Buchanan, 2019). These difficulties are enhanced by those encountered in the modelling of phase transitions that lead to cloud formation and convection, which are major sources of uncertainty in climate modelling (Bony et al., 2015). Machine learning techniques capable of learning geophysical flow dynamics would help improve those approximations and avoid running costly simulations resolving explicitly all spatial/temporal scales.

Recently, several efforts have been made to apply machine learning to the prediction of geophysical data (Wu et al., 2018), to learn parameterizations of subgrid processes in climate models (Krasnopolsky et al., 2005; Krasnopolsky and Fox-Rabinovitz, 2006; Rasp et al., 2018; Gentine et al., 2018; Brenowitz and Bretherton, 2018, 2019; Yuval and O'Gorman, 2020; Gettelman et al., 2021; Krasnopolsky

et al., 2013), to forecast (Liu et al., 2015; Grover et al., 2015; Haupt et al., 2018; Weyn et al., 2019) and nowcast (i.e. extremely short-term forecasting) weather variables (Xingjian et al., 2015; Shi et al., 2017; Sprenger et al., 2017), and to quantify the uncertainty of deterministic weather prediction (Scher and Messori, 2018). One of the greatest challenges is to replace equations of climate models with neural networks capable of producing reliable long- and short-term forecasts of meteorological variables. A first great step in this direction was the use of echo state networks (ESNs, Jaeger, 2001), a particular case of recurrent neural networks (RNNs), to forecast the behaviour of chaotic systems, such as the Lorenz (1963) and Kuramoto–Sivashinsky dynamics (Hyman and Nicolaenko, 1986). It was shown that ESN predictions of both systems attain performances comparable to those obtained with the exact equations (Pathak et al., 2017, 2018). Good performances were obtained adopting regularized ESNs in the short-term prediction of multidimensional chaotic time series, both from simulated and real data (Xu et al., 2018). This success motivated several follow-up studies with a focus on meteorological and climate data. These are based on the idea of feeding various statistical learning algorithms with data issued from dynamical systems of different complexity in order to study short-term predictability and capability of machine learning to reproduce long-term features of the input data dynamics. Recent examples include equation-informed moment matching for the Lorenz 1996 model (Lorenz, 1996; Schneider et al., 2017), multi-layer perceptrons to reanalysis data (Scher, 2018), or convolutional neural networks to simplified climate simulation models (Dueben and Bauer, 2018; Scher and Messori, 2019). All these learning algorithms were capable of providing some short-term predictability but failed at obtaining a long-term behaviour coherent with the input data.

The motivation for this study came from the evidence that a straightforward application of ESNs to high-dimensional geophysical data does not yield the same result quality obtained by Pathak et al. (2018) for the Lorenz 1963 and Kuramoto–Sivashinsky models. Here we will investigate the causes of this behaviour. Indeed, previous results (Scher, 2018; Dueben and Bauer, 2018; Scher and Messori, 2019) suggest that simulations of large-scale climate fields through deep-learning algorithms are not as straightforward as those of the chaotic systems considered by Pathak et al. (2018). We identify two main mechanisms responsible for these limitations: (i) the non-trivial interactions with small-scale motions carrying energy at large scales and (ii) the intermittent nature of the dynamics. Intermittency triggers large fluctuations of observables of the motion in time and space (Schertzer et al., 1997) and can result in non-smooth trajectories within the flow, leading to local unpredictability and increasing the number of degrees of freedom needed to describe the dynamics (Paladin and Vulpiani, 1987).

By applying ESNs to multiscale and intermittent systems, we investigate how scale separation improves ESN predictions. Our goal is to reproduce a surrogate of the large-scale dynamics of global sea-level pressure fields, a proxy of the atmospheric circulation. We begin by analysing three different dynamical systems: we simulate the effects of small scales by artificially introducing small-scale dynamics in the Lorenz 1963 equations (Lorenz, 1963) via additive noise, in the spirit of recent deep-learning studies with add-on stochastic components (Mukhin et al., 2015; Seleznev et al., 2019). We investigate the Pomeau–Manneville equations (Manneville, 1980) stochastically perturbed with additive noise to have an example of intermittent behaviour. We then analyse the performances of ESNs in the Lorenz 1996 system (Lorenz, 1996). The dynamics of this system is meant to mimic that of the atmospheric circulation, featuring both large-scale and small-scale variables with an intermittent behaviour. For all of those systems as well as for the sea-level pressure data, we show how the performances of ESNs in predicting the behaviour of the system deteriorate rapidly when small-scale dynamics feedback to large scales is important. The idea of using a moving average for scale separation is already established for meteorological variables (Eskridge et al., 1997). We choose the ESN framework following the results of Pathak et al. (2017, 2018) and an established literature about its ability to forecast chaotic time series and its stability to noise. For example, Shi and Han (2007) and Li et al. (2012) analyse and compare the predictive performances of simple and improved ESNs on simulated and observed 1D chaotic time series. We aim at understanding this sensitivity in a deeper way while assessing the possibility of reducing its impact on prediction through simple noise reduction methods.

The remainder of this article is organized as follows: in Sect. 2, we give an overview of the ESN method (Sect. 2.1), and then we introduce the metrics used to evaluate ESN performance (Sect. 2.2) and introduce the moving-average filter used to improve ESN performance (Sect. 2.3). Section 3 presents the results for each analysed system. First we show the results for the perturbed Lorenz 1963 equations, then for the Pomeau–Manneville intermittent map, and then for the Lorenz 1996 equations. Finally, we discuss the improvement in short-term prediction and the long-term attractor reconstruction obtained with the moving-average filter. We conclude by testing these ideas on atmospheric circulation data.

## 2 Methods

Reservoir computing is a variant of recurrent neural networks (RNNs) in which the input signal is connected to a fixed, randomly assigned dynamical system called a reservoir (Hinaut, 2013). The principle of reservoir computing first consists in projecting the input signal to a high-dimensional space in order to obtain a non-linear representation of the signal and then in performing a new projection between the high-dimensional space and the output units, usually via linear

regression or ridge regression. In our study, we use ESN, a particular case of RNN where the output and the input have the same dynamical form. In an ESN, neuron layers are replaced by a sparsely connected network (the reservoir), with randomly assigned fixed weights. We harvest reservoir states via a non-linear transform of the driving input and compute the output weights to create reservoir-to-output connections. The code is given in the Appendix, and it shows the parameters used for the computations.

We now briefly describe the ESN implementation. Vectors will be denoted in bold and matrices in upper case. Let $\boldsymbol{x}(t)$ be the $K$-dimensional observable consisting of $t = 1, 2, \ldots, T$ time iterations, originating from a dynamical system, and $\boldsymbol{r}(t)$ be the $N$-dimensional reservoir state; then,

$$\boldsymbol{r}(t + \mathrm{d}t) = \tanh(\mathbf{W}\boldsymbol{r}(t) + \mathbf{W}_{\mathrm{in}}\boldsymbol{x}(t)), \tag{1}$$

where $\mathbf{W}$ is the adjacency matrix of the reservoir: its dimensions are $N \times N$, and $N$ is the number of neurons in the reservoir. In ESNs, the neuron layers of classic deep neural networks are replaced by a single layer consisting of a sparsely connected random network, with coefficients uniformly distributed in $[-0.5; 0.5]$. The $N \times K$-dimensional matrix $\mathbf{W}_{\mathrm{in}}$ is the weight matrix of the connections between the input layer and the reservoir, and the coefficients are randomly sampled, as for $\mathbf{W}$. The output of the network at time step $t + \mathrm{d}t$ is

$$\mathbf{W}_{\mathrm{out}}\boldsymbol{r}(t + \mathrm{d}t) = \boldsymbol{y}(t + \mathrm{d}t), \tag{2}$$

where $\boldsymbol{y}(t + \mathrm{d}t)$ is the ESN prediction, and $\mathbf{W}_{\mathrm{out}}$ with dimensions $K \times N$ is the weight matrix of the connections between the reservoir neurons and the output layer. We estimate $\mathbf{W}_{\mathrm{out}}$ via a ridge regression (Hastie et al., 2015):

$$\mathbf{W}_{\mathrm{out}} = \boldsymbol{y}_{\mathrm{train}}\boldsymbol{r}^T[\boldsymbol{r}\boldsymbol{r}^T - \lambda I]^{-1}, \tag{3}$$

with $\lambda = 10^{-8}$ and $\boldsymbol{y}_{\mathrm{train}} \equiv \{\boldsymbol{y}(t) : (0 < t < T_{\mathrm{train}})\}$ as training datasets. Note that we have investigated different values of $\lambda$ spanning $10^{-8} < \lambda < 10^{-2}$ on the Lorenz 1963 example and found little improvement only when the network size was large, with $\lambda$ partially preventing overfitting. Values of $\lambda < 10^{-8}$ have not been investigated because they are too close to or below the numerical precision. In the prediction phase we have a recurrent relationship:

$$\boldsymbol{r}(t + \mathrm{d}t) = \tanh(W\boldsymbol{r}(t) + W_{\mathrm{in}}W_{\mathrm{out}}\boldsymbol{r}(t)). \tag{4}$$

## 2.1 ESN performance indicators

In this paper, we use three different indicators of performance of the ESN: a statistical distributional test to measure how the distributions of observables derived from ESN match those of the target data, a predictability horizon test and the initial forecast error. They are described below.

### 2.1.1 Statistical distributional test

As a first diagnostic of the performances of ESNs, we aim at assessing whether the marginal distribution of the forecast values for a given dynamical system is significantly different from the invariant distribution of the system itself. To this purpose, we conduct a $\chi^2$ test (Cochran, 1952), designed as follows. Let $U$ be a system observable, linked to the original variables of the systems via a function $\zeta$, a function mapping between two spaces, such that $u(t) = \zeta(\boldsymbol{x}(t))$ with support $R_U$ and probability density function $f_U(u)$, and let $u(t)$ be a sample trajectory from $U$. Note that $u(t)$ does not correspond to $x(t)$; it is constructed using the observable output of the dynamical system. Let now $\hat{f}_U(u)$ be an approximation of $f_U(u)$, namely the histogram of $u$ over $i = 1, \ldots, M$ bins. Note that, if $u$ spans the entire phase space, $\hat{f}_U(u)$ is the numerical approximation of the Sinai–Ruelle–Bowen measure of the dynamical system (Eckmann and Ruelle, 1985; Young, 2002). Let now $V$ be the variable generated by the ESN forecasting, with support $R_V = R_U$, $v(t)$ the forecast sample, $g_V(v)$ its probability density function and $\hat{g}_V(v)$ the histogram of the forecast sample. Formally, $R_U$ and $R_V$ are Banach spaces, whose dimension depends on the choice of the function $\zeta$. For example, we will use as $\zeta$s one of the variables of the system or the sum of all the variables. We test the null hypothesis that the marginal distribution of the forecast sample is the same as the invariant distribution of the system against the alternative hypothesis that the two distributions are significantly different.

$H_0 :$ $f_U(u) = g_V(v)$ for every $u \in R_U$

$H_1 :$ $f_U(u) \neq g_V(v)$ for any $u \in R_U$

Under $H_0$, $\hat{f}_U(u)$ is the expected value for $\hat{g}_V(v)$, which implies that observed differences $(\hat{g}_V(v) - \hat{f}_U(u))$ are due to random errors and are then independent and identically distributed Gaussian random variables. Statistical theory shows that, given $H_0$ is true, the test statistics

$$\Sigma = \sum_{i=1}^{M} \frac{(\hat{g}_V^i(v) - \hat{f}_U^i(u))^2}{\hat{f}_U^i(u)} \tag{5}$$

is distributed as a $\chi^2$ random variable with $M$ degrees of freedom, $\chi^2(M)$. Then, to test the null hypothesis at the level $\alpha$, the observed value of the test statistics $\Sigma$ is compared to the critical value corresponding to the $1 - \alpha$ quantile of the $\chi^2$ distribution, $\Sigma_c = \chi_{1-\alpha}^2(M)$: if $\Sigma > \Sigma_c$, the null hypothesis must be rejected in favour of the specified alternative. Since we are evaluating the proximity between distributions, the Kullback–Leibler (KL) divergence could be considered a more natural measure. However, we decide to rely on the $\chi^2$ test because of its feasibility while maintaining some equivalence to KL. In fact, both the KL and $\chi^2$ are non-symmetric statistical divergences, an ideal property when measuring a proximity to a reference probability distribution (Karagrigoriou, 2012). It has been known for a long time (Berkson

et al., 1980) that statistical estimation based on minimum $\chi^2$ is equivalent to most other objective functions, including the maximum likelihood and, thus, the minimum KL. In fact, the KL divergence is linked to maximum likelihood both in an estimation and in a testing setting. In point estimation, maximum likelihood parameter estimates minimize the KL divergence from the chosen statistical model; in hypothesis testing, the KL divergence can be used to quantify the loss of power of likelihood ratio tests in case of model misspecification (Eguchi and Copas, 2006). On the other hand, in contrast to the KL, the $\chi^2$ divergence has the advantage of converging to a known distribution under the null hypothesis, independently of the parametric form of the reference distribution.

In our setup, we encounter two limitations in using the standard $\chi^2$ test. First, problems may arise when $\hat{f}_U(u)$, i.e. if the sample distribution does not span the entire support of the invariant distribution of the system. We observe this in a relatively small number of cases; since aggregating the bins would introduce unwanted complications, we decide to discard the pathological cases, controlling the effect empirically as described below. Moreover, even producing relatively large samples, we are not able to actually observe the invariant distribution of the considered system, which would require much longer simulations. As a consequence, we would observe excessive rejection rates when testing samples generated under $H_0$.

We decide to control these two effects by using a Monte Carlo approach. To this purpose, we generate $10^5$ samples $u(t) = \zeta(\boldsymbol{x}(t))$ under the null hypothesis, and we compute the test statistic for each one according to Eq. (5). Then, we use the $(1 - \alpha)$ quantile of the empirical distribution of $\Sigma$ – instead of the theoretical $\chi^2(M)$ – to determine the critical threshold $\Sigma_c$. As a last remark, we notice that we are making inferences in repeated test settings, as the performance of the ESN is tested $10^5$ times. Performing a high number of independent tests at a chosen level $\alpha$ increases the observed rejection rate: in fact, even if the samples are drawn under $H_0$, extreme events become more likely, resulting in an increased probability of erroneously rejecting the null hypothesis. To avoid this problem, we apply the Bonferroni correction (Bonferroni, 1936), testing each one of the $m = 10^5$ available samples at the level $\alpha' = \frac{\alpha}{m}$, with $\alpha = 0.05$. Averaging the test results over several sample pairs $u(t)$, $v(t)$, we obtain a rejection rate of $0 < \phi < 1$ that we use to measure the adherence of an ESN trajectory $v(t)$ to trajectories obtained via the equations. If $\phi = 0$, almost all the ESN trajectories can shadow original trajectories; if $\phi = 1$, none of the ESN trajectories resemble those of the systems of equations.

### 2.1.2   Predictability horizon

As a measure of the predictability horizon of the ESN forecast compared to the equations, we use the absolute predic-

tion error (APE),

$$\text{APE}(t) = |u(t) - v(t)|, \tag{6}$$

and we define the predictability horizon $\tau_s$ as the first time that APE exceeds a certain threshold $s$:

$$\tau_s = \inf\{t, \text{APE}(t) > s\}. \tag{7}$$

Indeed, APE can equivalently be written as $\Delta t \dot{u}$. We link $s$ to the average separation of observations in the observable $u$ and we fix

$$s = \frac{1}{T-1} \sum_{t=2}^{T-1} [u(t) - u(t-1)]. \tag{8}$$

We have tested the sensitivity of results against the exact definition of $s$. We interpret $\tau_s$ as a natural measure of the Lyapunov time $\vartheta$, namely the time it takes for an ensemble of trajectories of a dynamical system to diverge (Faranda et al., 2012; Panichi and Turchetti, 2018).

### 2.1.3   Initial forecast error

The initial error is given by $\eta = \text{APE}(t = 1)$, for the first time step after the initial condition at $t = 0$. We expect $\eta$ to reduce as the training time increases.

## 2.2   Moving-average filter

Equipped with these indicators, we analyse two sets of simulations performed with and without smoothing, which was implemented using a moving-average filter. The moving-average operation is the integral of $u(t)$ between $t$ and $t - w$, where $w$ is the window size of the moving average. The simple moving-average filter can be seen as a non-parametric time series smoother (see e.g. Brockwell and Davis, 2016, chapter 1.5). It can be applied to smooth out (relatively) high frequencies in a time series, both to de-noise the observations of a process and to estimate trend-cycle components, if present. Moving averaging consists, in practice, in replacing the trajectory $\boldsymbol{x}(t)$ by a value $\boldsymbol{x}^{(f)}(t)$, obtained by averaging the previous $w$ observations. If the time dimension is discrete (like in the Pomeau–Manneville system), it is defined as

$$\boldsymbol{x}^{(f)}(t) = \frac{1}{w} \sum_{i=0}^{w-1} \boldsymbol{x}(t-i), \tag{8}$$

while for continuous time systems (like the Lorenz 1963 system), the sum is formally replaced by an integral:

$$\boldsymbol{x}^{(f)}(t) = \frac{1}{w} \int_{t}^{t+w} \boldsymbol{x}(\varsigma) \mathrm{d}\varsigma. \tag{9}$$

We can define the residuals as

$$\delta \boldsymbol{x}(t) = \boldsymbol{x}^{(f)}(t) - \boldsymbol{x}(t). \tag{10}$$

In practice, the computation always refers to the discrete time case, as continuous time systems are also sampled at finite time steps. Since echo state networks are known to be sensitive to noise (see e.g. Shi and Han, 2007), we exploit the simple moving-average filter to smooth out high-frequency noise and assess the results for different smoothing windows $w$. We find that the choice of the moving-average window $w$ must respect two conditions: it should be large enough to smooth out the noise but smaller than the characteristic time $\tau$ of the large-scale fluctuations of the system. For chaotic systems, $\tau$ can be derived knowing the rate of exponential divergence of the trajectories, a quantity linked to the Lyapunov exponents (Wolf et al., 1985), and $\tau$ is known as the Lyapunov time.

We also note that we can express explicitly the original variables $\boldsymbol{x}(t)$ as a function of the filtered variables $\boldsymbol{x}^{(f)}(t)$ as

$$\boldsymbol{x}(t) = w(\boldsymbol{x}^{(f)}(t) - \boldsymbol{x}^{(f)}(t-1)) + \boldsymbol{x}(t-w). \tag{11}$$

We will test this formula for stochastically perturbed systems to evaluate the error introduced by the use of residuals $\delta\boldsymbol{x}$.

### 2.3 Testing ESN on filtered dynamics

Here we describe the algorithm used to test ESN performance on filtered dynamics.

1. Simulate the reference trajectory $\boldsymbol{x}(t)$ using the equations of the dynamical systems and standardize $\boldsymbol{x}(t)$ by subtracting the mean and dividing by its standard deviation.

2. Perform the moving-average filter to obtain $\boldsymbol{x}^{(f)}(t)$.

3. Extract from $\boldsymbol{x}^{(f)}(t)$ a training set $\boldsymbol{x}^{(f)}_{\text{train}}(t) \equiv \{\boldsymbol{x}(t) : (0 < t < T_{\text{train}})\}$.

4. Train the ESN on the $\boldsymbol{x}^{(f)}_{\text{train}}(t)$ dataset.

5. Obtain the ESN forecast $\boldsymbol{y}^{(f)}(t)$ for $T_{\text{train}} < t < T$. Note that the relation between $\boldsymbol{y}(t)$ and $\boldsymbol{x}(t)$ is given in Eqs. (1)–(2).

6. Add residuals (Eq. 10) to the $\boldsymbol{y}^{(f)}(t)$ sample as $\boldsymbol{y}(t) = \boldsymbol{y}^{(f)}(t) + \delta\boldsymbol{x}$, where $\delta\boldsymbol{x}$ is randomly sampled from the $\delta\boldsymbol{x}^{(f)}_{\text{train}}(t)$.

7. Compute the observables $v(t) = \zeta(\boldsymbol{y}(t))$ and $u(t) = \zeta(\boldsymbol{x}(T_{\text{train}} < t < T)$. Note that $\boldsymbol{x}(T_{\text{train}} < t < T)$ is the ensemble of *true values* of the original dynamical system.

8. Using $u(t)$ and $v(t)$, compute the metrics $\phi$, $\tau$ and $\eta$ and evaluate the forecasts.

As an alternative to step 6, one can also use Eq. (11) and obtain

$$v(t) \simeq w(v^{(f)}(t) - v^{(f)}(t-1)) + v(t-w), \tag{12}$$

which does not require the use of residuals $\delta\boldsymbol{x}(t)$. Here $v^{(f)}(t) = \zeta(\boldsymbol{y}^{(f)}(t)))$. The latter equation is only approximate because of the moving-average filter.

## 3 Results

The systems we analyse are the Lorenz 1963 attractor (Lorenz, 1963) with the classical parameters, discretized with an Euler scheme and $dt = 0.001$, the Pomeau–Manneville intermittent map (Manneville, 1980), the Lorenz 1996 equations (Lorenz, 1996) and the NCEP sea-level pressure data (Saha et al., 2014).

### 3.1 Lorenz 1963 equations

The Lorenz system is a simplified model of Rayleigh–Benard convection, derived by Edward Northon Lorenz (Lorenz, 1963). It is an autonomous continuous dynamical system with three variables $\{x, y, z\}$ parameterizing respectively the convective motion, the horizontal temperature gradient and the vertical temperature gradient. It is written as

$$\frac{dx}{dt} = \sigma(y - x) + \epsilon\xi_x(t),$$
$$\frac{dy}{dt} = -xz + \varrho x - y + \epsilon\xi_y(t),$$
$$\frac{dz}{dt} = xy - bz + \epsilon\xi_z(t), \tag{13}$$

where $\sigma$, $\varrho$ and $b$ are three parameters, $\sigma$ mimicking the Prandtl number, $\varrho$ the reduced Rayleigh number and $b$ the geometry of convection cells. The Lorenz model is usually defined using Eq. (13), with $\sigma = 10$, $\varrho = 28$ and $b = 8/3$. A deterministic trajectory of the system is shown in Fig. 1a. It has been obtained by integrating numerically the Lorenz equations with an Euler scheme ($dt = 0.001$). We are aware that an advanced time stepper (e.g. Runge–Kutta) would provide better accuracy. However, when considering daily or 6-hourly data, as commonly done in climate sciences and analyses, we hardly are in the case of a smooth time stepper. We therefore stick to the Euler method for similarity to the climate data used in the last section of the paper. The system is perturbed via additive noise: $\xi_x(t)$, $\xi_y(t)$ and $\xi_z(t)$ are independent and identically distributed (i.i.d.) random variables all drawn from a Gaussian distribution. The initial conditions are randomly selected within a long trajectory of $5 \times 10^6$ iterations. First, we study the dependence of the ESN on the training length in the deterministic system ($\epsilon = 0$, Fig. 1b–d). We analyse the behaviour of the rejection rate $\phi$ (panel b), the predictability horizon $\tau_s$ (panel c) and the initial error $\eta$ (panel d) as a function of the training sample size. Our

analysis suggests that $t \sim 10^2$ is a minimum sufficient choice for the training window. We compare this time to the typical timescales of the motion of the systems, determined via the maximum Lyapunov exponent $\lambda$. For the Lorenz 1963 system, $\lambda = 0.9$, so that the Lyapunov time $\vartheta \approx \mathcal{O}\left(\frac{1}{\lambda}\right) \approx 1.1$. From the previous analysis we should train the network at least for $t > 100\vartheta$. For the other systems analysed in this article, we take this condition as a lower boundary for the training times.

To exemplify the effectiveness of the moving-average filter in improving the machine learning performances, in Fig. 2 we show 10 ESN trajectories obtained without a moving average (black) and with (red) a moving-average window $w = 0.01$ and compare them to the reference trajectory (blue) obtained with $\epsilon = 0.1$. The value of $w = 10\mathrm{d}t = 0.01$ respects the condition $w \ll \vartheta$. Indeed, the APE averaged over the two groups of trajectories (Fig. 2b) shows an evident gain in accuracy (a factor of $\sim 10$) when the moving-average procedure is applied. We now study in a more systematic way the dependence of the ESN performance on noise intensity $\epsilon$, network size $N$ and for three different averaging windows $w = 0$, $w = 0.01$, and $w = 0.05$. We produce, for each combination, 100 ESN forecasts. Figure 3 shows $\phi$ (a), $\log(\tau_{s=1})$ (b) and $\log(\eta)$ (c) computed setting the $u \equiv x$ variable of the Lorenz 1963 system (results qualitatively do not depend on the chosen variable). In each panel from left to right the moving-average window is increasing; upper sub-panels are obtained using the exact expression in Eq. (12) and lower panels using the residuals in Eq. (10). For increasing noise intensity and for small reservoir sizes, the performances without moving averages (left sub-panels) rapidly get worse. The moving-average smoothing with $w = 0.01$ (central sub-panels) improves the performance for $\log(\tau_{s=1})$ (b) and $\log(\eta)$ (c), except when the noise is too large ($\epsilon = 1$). Hereafter we denote with log the natural logarithm. When the moving-average window is too large (right panels), the performances of $\phi$ decrease. This failure can be attributed to the fact that residuals $\delta x$ (Eq. 10) are of the same order of magnitude of the ESN-predicted fields for large $\epsilon$. Indeed, if we use the formula provided in Eq. (12) as an alternative to step 6, we can evaluate the error introduced in the residuals. The results shown in Fig. 3 suggest that residuals can be used without problems when the noise is small compared with the dynamics. When $\epsilon$ is close to 1, the residuals overlay the deterministic dynamics and ESN forecasts are poor. In this case, the exact formulation in Eq. (12) appears much better.

## 3.2 Pomeau–Manneville intermittent map

Several dynamical systems, including the Earth's climate, display intermittency; i.e. the time series of a variable issued by the system can experience sudden chaotic fluctuations as well as a predictable behaviour where the observables have small fluctuations. In atmospheric dynamics, such behaviour is observed in the switching between zonal and meridional phases of the mid-latitude dynamics if a time series of the wind speed at one location is observed: when a cyclonic structure passes through the area, the wind has high values and large fluctuations, and when an anticyclonic structure is present, the wind is low and fluctuations are smaller (Weeks et al., 1997; Faranda et al., 2016). It is then of practical interest to study the performance of ESNs in Pomeau–Manneville predictions as they are a first prototypical example of the intermittent behaviour found in climate data.

In particular, the Pomeau–Manneville (Manneville, 1980) map is probably the simplest example of intermittent behaviour, produced by a 1D discrete deterministic map given by

$$x_{t+1} = \mathrm{mod}(x_t + x_t^{1+a}, 1) + \epsilon\xi(t), \qquad (14)$$

where $0 < a < 1$ is a parameter. We use $a = 0.91$ in this study and a trajectory consisting of $5 \times 10^5$ iterations. The system is perturbed via additive noise $\xi(t)$ drawn from a Gaussian distribution, and initial conditions are randomly drawn by a long trajectory, as for the Lorenz 1963 system. It is well known that Pomeau–Manneville systems exhibit sub-exponential separation of nearby trajectories, and then the Lyapunov exponent is $\lambda = 0$. However, one can define a Lyapunov exponent for the non-ergodic phase of the dynamics and extract a characteristic timescale (Korabel and Barkai, 2009). From this latter reference, we can derive a value $\lambda \simeq 0.2$ for $a = 0.91$, implying $w < \tau \simeq 5$. For the Pomeau–Manneville map, we set $u(t) \equiv x(t)$. We find that the best matches between ESNs and equations in terms of the $\phi$ indicator are obtained for $w = 3$.

Results for the Pomeau–Manneville map are shown in Fig. 4. We first observe that the ESN forecast of the intermittent dynamics of the Pomeau–Manneville map is much more challenging than for the Lorenz system as a consequence of the deterministic intermittent behaviour of this system. For the simulations performed with $w = 0$, the ESN cannot simulate an intermittent behaviour, for all noise intensities and reservoir sizes. This is reflected in the behaviour of the indicators. In the deterministic limit, the ESN fails to reproduce the invariant density in 80 % of the cases ($\phi \simeq 0.8$). We can therefore speculate that there is an intrinsic problem in reproducing intermittency driven by the deterministic chaos. For intermediate noise intensities, $\phi > 0.9$ (Fig. 4a). The predictability horizon $\log(\tau_{s=0.5})$ for the short-term forecast is small (Fig. 4d) and the initial error large (Fig. 4g). It appears that in smaller networks, the ESN keeps better track of the initial conditions, so that the ensemble shows smaller divergences $\log(\tau_{s=0.5})$. The moving-average procedure with $w = 3$ partially improves the performances (Fig. 4b, c, e, f, h, i), and it enables ESNs to simulate an intermittent behaviour (Fig. 5). Performances are again better when using the exact formula in Eq. (12) (Fig. 4b, e, h) than using the residuals $\delta x$ (Fig. 4c, f, i). Figure 5a shows the intermittent behaviour of

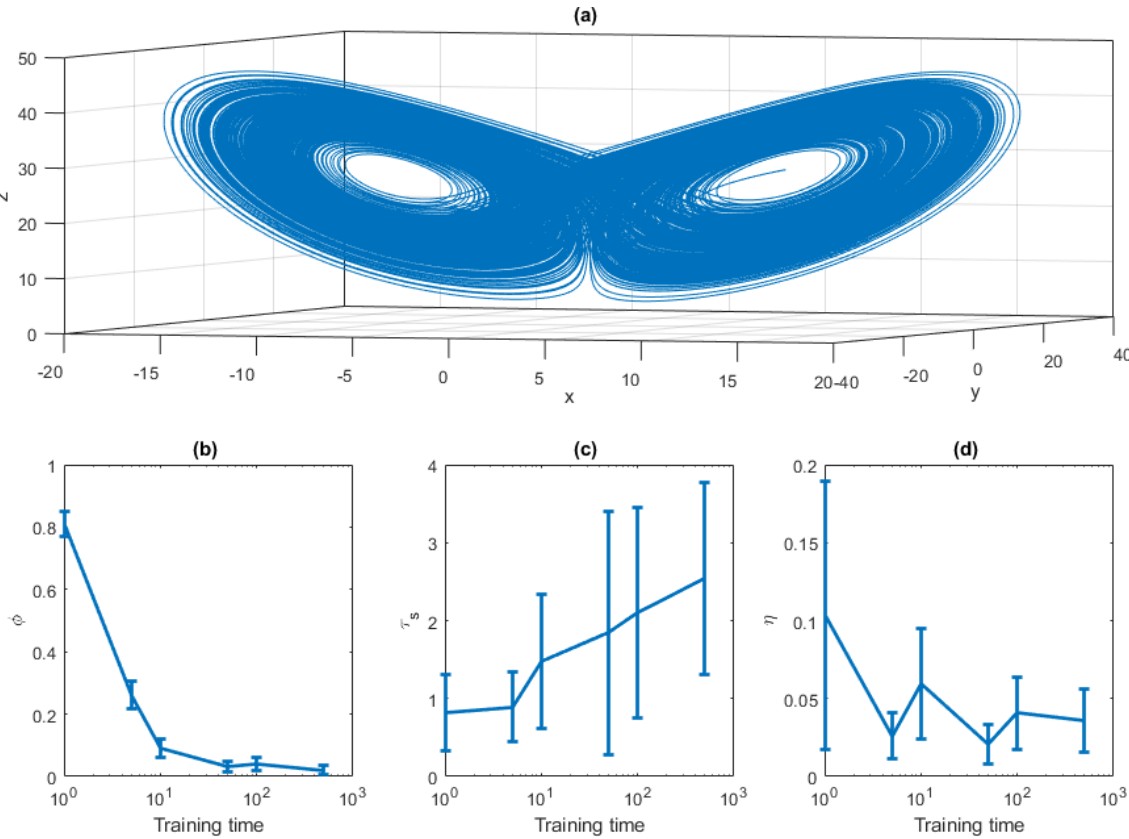

**Figure 1. (a)** Lorenz 1963 attractor obtained with an Euler scheme with $dt = 0.001$, $\sigma = 10$, $r = 28$ and $b = 8/3$. Panels **(b)**–**(d)** show the performance indicator as a function of the training time. **(b)** The rejection rate $\phi$ of the invariant density test for the $x$ variable; **(c)** the first time $t$ such that APE > 1; **(d)** the initial error $\eta$. The error bar represents the average and the standard deviation of the mean over 100 realizations.

the data generated with the ESN trained on moving-average data of the Pomeau–Manneville system (red) and compared to the target time series (blue). ESN simulations do not reproduce the intermittency in the average of the target signal, which shifts from $x \sim 0$ in the non-intermittent phase to $0.2 < x < 1$ in the intermittent phase. ESN simulations only show some second-order intermittency in the fluctuations while keeping a constant average. Figure 5b displays the power spectra showing in both cases a power law decay, which is typical of turbulent phenomena. Although the intermittent behaviour is captured, this realization of an ESN shows that the values are concentrated around $x = 0.5$ for the ESN prediction, whereas the non-intermittent phase peaks around $x = 0$ for the target data.

## 3.3 The Lorenz 1996 system

Before running the ESN algorithm on actual climate data, we test our idea in a more sophisticated, and yet still idealized, model of atmospheric dynamics, namely the Lorenz 1996 equations (Lorenz, 1996). This model explicitly separates two scales and therefore will provide a good test for our ESN algorithm. The Lorenz 1996 system consists of a lattice

of large-scale resolved variables $X$, coupled to small-scale variables $Y$, whose dynamics can be intermittent. The model is defined via two sets of equations:

$$\frac{dX_i}{dt} = X_{i-1}(X_{i+1} - X_{i-2}) - X_i + F - \frac{hc}{b}\sum_{j=1}^{J}Y_{j,i}, \quad (15)$$

$$\frac{dY_{j,i}}{dt} = cbY_{j+1,i}(Y_{j-1,i} - Y_{j+2,i}) - cY_{j,i} + \frac{hc}{b}X_i, \quad (16)$$

where $i = 1, \ldots, I$ and $j = 1, 2, \ldots, J$ denote respectively the number of large-scale $X$ and small-scale $Y$ variables. Large-scale variables are meant to represent the meanders of the jet stream driving the weather at mid latitudes. The first term on the right-hand side represents advection, the second diffusion, while $F$ mimics an external forcing. The system is controlled via the parameters $b$ and $c$ (the timescale of the fast variables compared to the small variables) and via $h$ (the coupling between large and small scales). From now on, we fix $I = 30$, $J = 5$ and $F = b = 10$ as these parameters are typically used to explore the behaviour of the system (Frank et al., 2014). We integrate the equations with an Euler scheme ($dt = 10^{-3}$) from the initial conditions $Y_{j,i} = X_i = F$, where

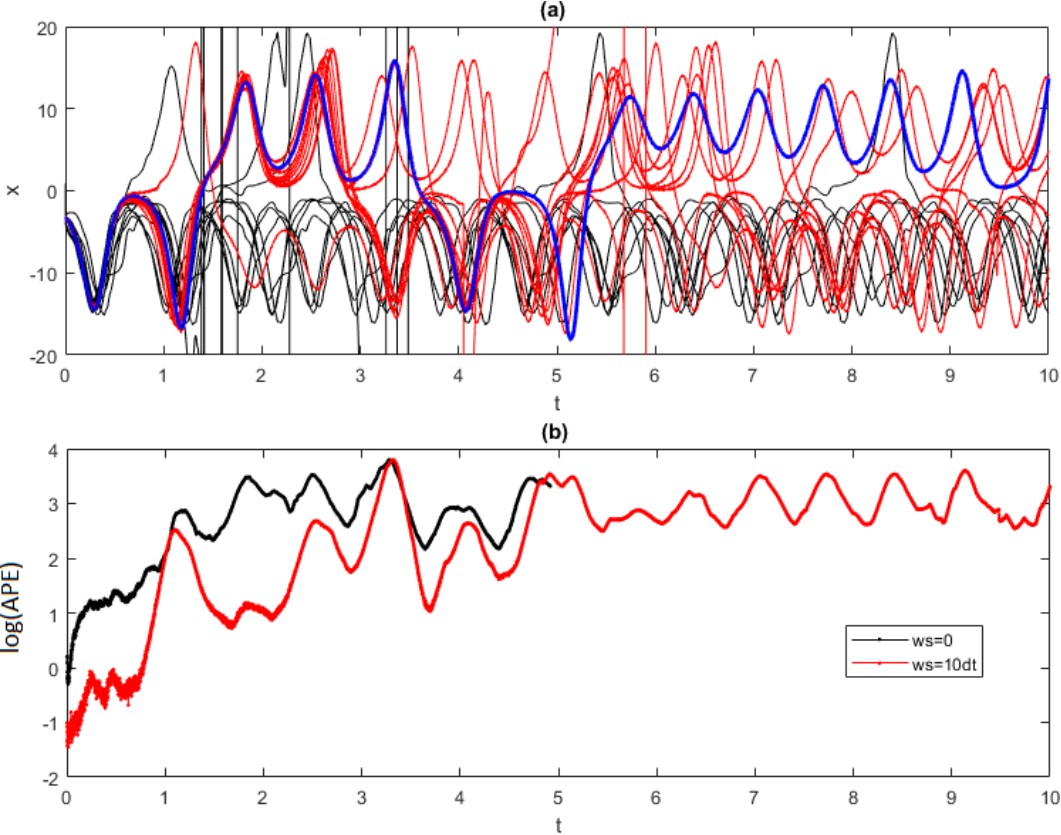

**Figure 2. (a)** Trajectories predicted using ESN on the Lorenz 1963 attractor for the variable $x$. The attractor is perturbed with Gaussian noise with variance $\epsilon = 0.1$. The target trajectory is shown in blue. Ten trajectories obtained without a moving average (black) show an earlier divergence compared to 10 trajectories where the moving average is performed with a window size of $w = 10\mathrm{d}t = 0.01$ (red). Panel **(b)** shows the evolution of the log(APE) averaged over the trajectories for the cases with $w = 0.01$ (red) and $w = 0$ (black). The trajectories are all obtained after training the ESN for the same training set consisting of a trajectory of $10^5$ time steps. Each trajectory consists of $10^4$ time steps.

only one mode is perturbed as $X_{i=1} = F + \varepsilon$ and $Y_{j,i=1} = F + \varepsilon^2$. Here $\varepsilon = 10^{-3}$. We discard about $2 \times 10^3$ iterations to reach a stationary state on the attractor, and we retain $5 \times 10^4$ iterations. When $c$ and $h$ vary, different interactions between large and small scales can be achieved. A few examples of simulations of the first mode $X_1$ and $Y_1$ are given in Fig. 6. Figure 6a, c show simulations obtained for $h = 1$ by varying $c$: the larger the $c$, the more intermittent the behaviour of the fast scales. Figure 6b, d show simulations obtained for different coupling $h$ at fixed $c = 10$: when $h = 0$, there is no small-scale dynamics.

For the Lorenz 1996 model, we do not need to apply a moving-average filter to the data, as we can train the ESN on the large-scale variables only. Indeed, we can explore what happens to the ESN performances if we turn on and off intermittency and/or the small- to large-scale coupling, without introducing any additional noise term. Moreover, we can also learn the Lorenz 1996 dynamics on the $X$ variables only or learn the dynamics on both $X$ and $Y$ variables. The purpose of this analysis is to assess whether the ESNs are capable of learning the dynamics of the large-scale variables $X$ alone and how this capability is influenced by the coupling and the intermittency of the small-scale variables $Y$. Using the same simulations presented in Fig. 6, we train the ESN on the first $2.5 \times 10^4$ iterations, and then perform, changing the initial conditions to 100 different ESN predictions for $2.5 \times 10^4$ more iterations. We apply our performance indicators not to the entire $I$-dimensional $X$ variable $(X_1, \ldots, X_I)$, as the $\chi^2$ test becomes intractable in high dimensions, but rather to the average of the large-scale variables $X$. Consistently with our notation, it means that $u(t) \equiv \sum_{i=1}^{I} X_i(t)$. The behaviour of each variable $X_i$ is similar, so the average is representative of the collective behaviour. The rate of failure $\phi$ is very high (not shown) because even when the dynamics is well captured by the ESN in terms of characteristic timescales and spatial scales, the predicted variables are not scaled and centred like those of the original systems. For the following analysis, we therefore replace $\phi$ with the $\chi^2$ distance $\Sigma$ (Eq. 5). The use of $\Sigma$ allows for better highlighting of the differences in the ESN performance with respect to the chosen param-

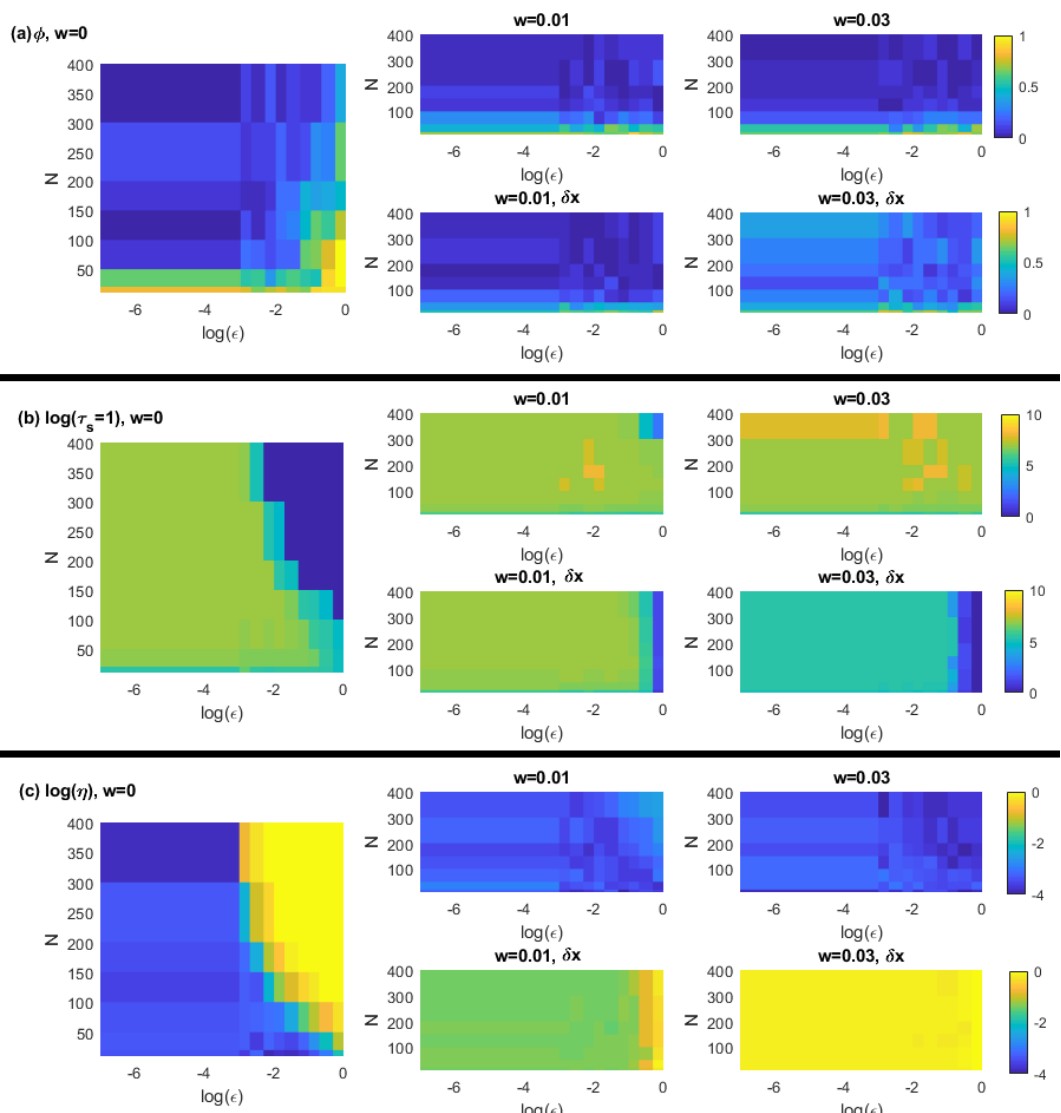

**Figure 3.** Lorenz 1963 analysis for increasing noise intensity $\epsilon$ ($x$ axes) and number of neurons $N$ ($y$ axes). The colour scale represents $\phi$ the rate of failure of the $\chi^2$ test (size $\alpha = 0.05$) **(a)**; the logarithm of predictability horizon $\log(\tau_{s=1})$ **(b)**; the logarithm of initial error $\log(\eta)$ **(c)**. These diagnostics have been computed on the observable $u(t) \equiv x(t)$. All the values are averages over 30 realizations. Left-hand-side sub-panels refer to results without moving averages, central sub-panels with averaging window $w = 0.01$, and right-hand-side panels with averaging window $w = 0.03$. Upper sub-panels are obtained using the exact expression in Eq. (12) and lower sub-panels using the residuals in Eq. (10). The trajectories are all obtained after training the ESN for $10^5$ time steps. Each trajectory consists of $10^4$ time steps.

eters. The same considerations also apply to the analysis of the sea-level pressure data reported in the next paragraph.

Results of the ESN simulations for the Lorenz 1996 system are reported in Fig. 7. In Fig. 7a, c, e, ESN predictions are obtained by varying $c$ at fixed $h = 1$ and in Fig. 7b, d, f by varying $h$ at fixed $c = 10$. The continuous lines refer to results obtained by feeding the ESN with only the $X$ variables, dotted lines with both $X$ and $Y$. For the $\chi^2$ distance $\Sigma$ (Fig. 7a, b), performances show a large dependence on both intermittency $c$ and coupling $h$. First of all, we remark that learning both $X$ and $Y$ variables leads to higher distances $\Sigma$,

except for the non-intermittent case, $c = 1$. For $c > 1$, the dynamics learnt on both $X$ and $Y$ never settles on a stationary state resembling that of the Lorenz 1996 model. When $c > 1$ and only the dynamics of the $X$ variables is learnt, the dependence on $N$ when $h$ is varied is non-monotonic, and better performances are achieved for $800 < N < 1200$. For this range, the dynamics settles on stationary states whose spatiotemporal evolution resembles that of the Lorenz 1996 model, although the variability of timescales and spatial scales is different from the target. An example is provided in Fig. 8 for $N = 800$. This figure shows an average example of the per-

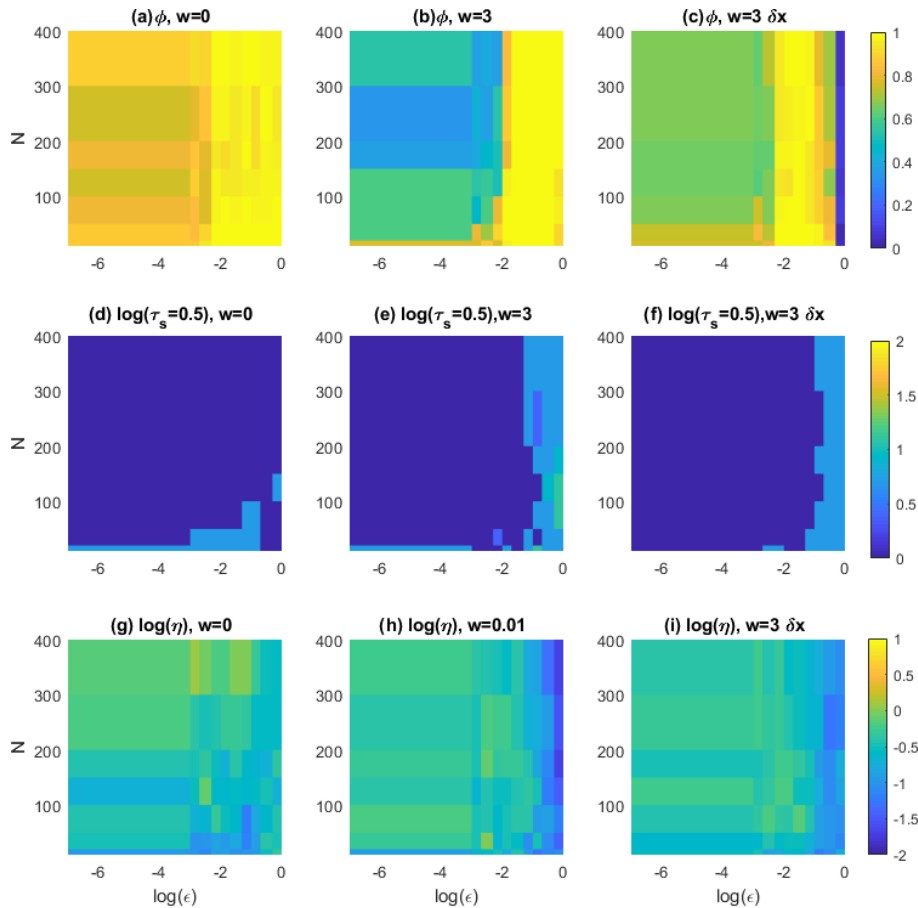

**Figure 4.** Analysis of the Pomeau–Manneville system for increasing noise intensity $\epsilon$ ($x$ axes) and number of neurons $N$ ($y$ axes). The colour scale represents $\phi$ the rate of failure of the $\chi^2$ test (size $\alpha = 0.05$) **(a–c)**; the logarithm of the predictability horizon $\log(\tau_{s=0.5})$ **(d–f)**; the logarithm of initial error $\log(\eta)$ **(g–i)**. These diagnostics have been computed on the observable $u(t) \equiv x(t)$. All the values are averages over 30 realizations. Panels **(a)**, **(d)**, and **(g)** refer to results without moving averages, **(b)**, **(c)**, **(e)**, **(f)**, **(h)**, and **(i)** with averaging window $w = 3$, **(c)**, **(f)**, and **(i)**. Panels **(b)**, **(e)**, and **(h)** are obtained using the exact expression in Eq. (12) and **(c)**, **(f)**, and **(i)** using the residuals $\delta x$ in Eq. (10). The trajectories are all obtained after training the ESN for $10^5$ time steps. Each trajectory consists of $10^4$ time steps.

formances of ESNs in reproducing the Lorenz 1996 system when the fit succeeds. For comparison, we refer to the results by Vlachas et al. (2020), which show that better fits of the Lorenz 1996 dynamics can be obtained using back-propagation algorithms.

Let us now analyse the two indicators of short-term forecasts. Figure 7c, d display the predictability horizon $\tau_s$ with $s = 1$. The best performances are achieved for the non-intermittent case $c = 1$ and learning both $X$ and $Y$. When only $X$ is learnt, we again get better performances in terms of $\tau_s$ for rather small network sizes. The performances for $c > 1$ are better when only $X$ variables are learnt. The good performances of ESNs in learning only the large-scale variables $X$ are even more surprising when looking at initial error $\eta$ (Fig. 7), which is 1 order of magnitude smaller when $X$ and $Y$ are learnt. Despite this advantage in the initial conditions, the ESN performances on $(X, Y)$ are better only when the dynamics of $Y$ is non-intermittent. We find clear indica-

tions that large intermittency ($c = 25$) and strong small- to large-scale variables coupling ($h = 1$) worsen the ESN performances, supporting the claims made for the Lorenz 1963 and Pomeau–Manneville systems.

### 3.4 The NCEP sea-level pressure data

We now test the effectiveness of the moving-average procedure in learning the behaviour of multiscale and intermittent systems on climate data issued by reanalysis projects. We use data from the National Centers for Environmental Prediction (NCEP) version 2 (Saha et al., 2014) with a horizontal resolution of 2.5°. We adopt the global 6-hourly sea-level pressure (SLP) field from 1979 to 31 August 2019 TS1 as the meteorological variable proxy for the atmospheric circulation. It traces cyclones (anticyclones) with minima (maxima) of the SLP fields. The major modes of variability affecting mid-latitude weather are often defined in terms of the empirical

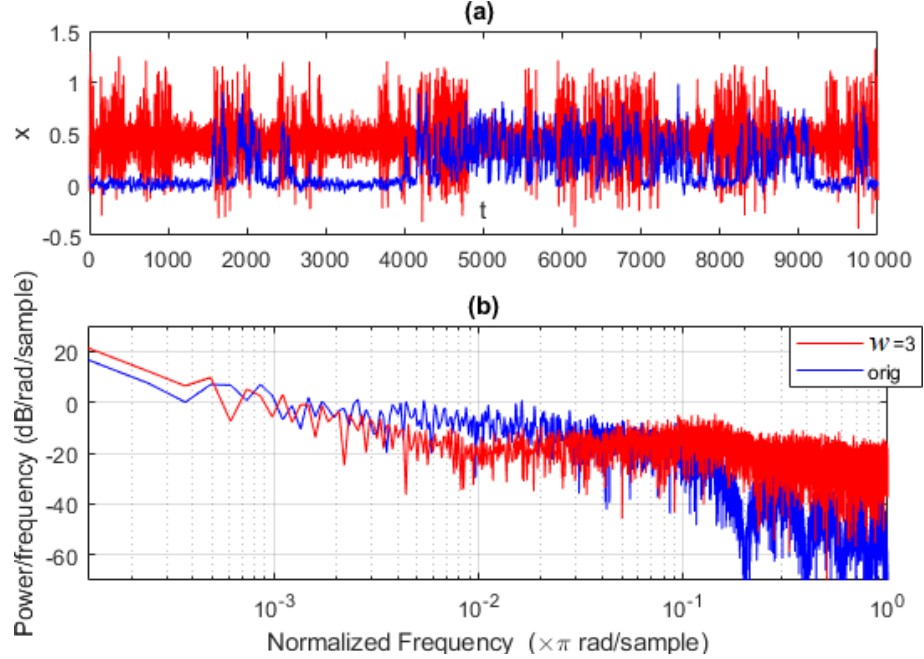

**Figure 5.** Pomeau–Manneville ESN simulation (red) showing an intermittent behaviour and compared to the target trajectory (blue). The ESN trajectory is obtained after training the ESN for $10^5$ time steps using the moving-average time series with $w = 3$. It consists of $10^4$ time steps. Cases $w = 0$ are not shown as trajectories always diverge. Evolution of trajectories in time **(a)** and Fourier power spectra **(b)**.

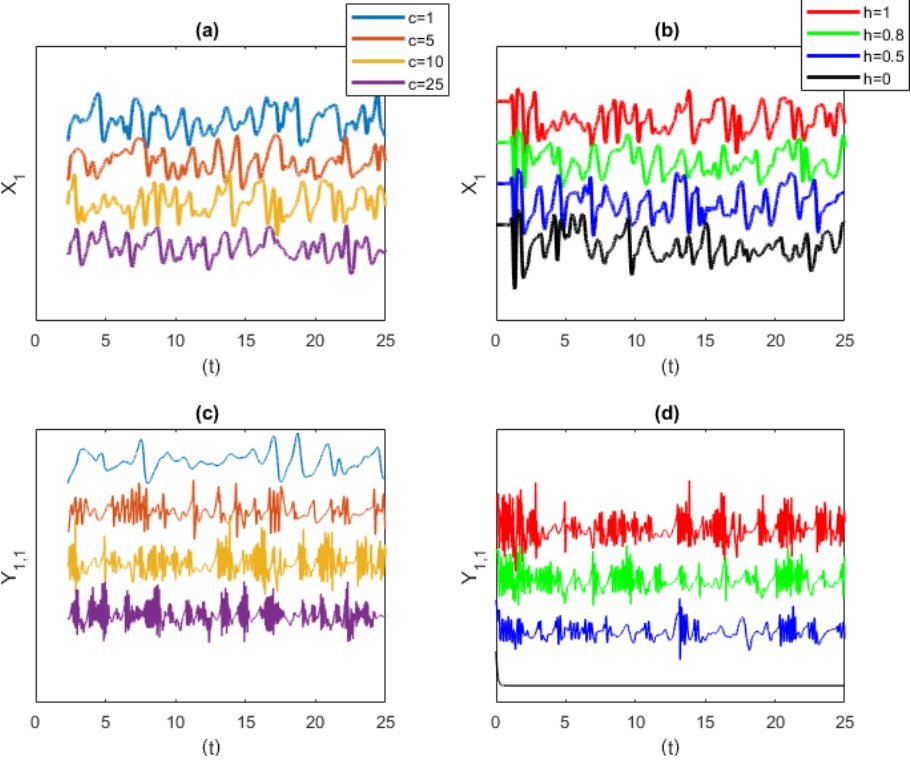

**Figure 6.** Lorenz 1996 simulations for the large-scale variable $X_1$ **(a, b)** and small-scale variable $Y_{1,1}$ **(c, d)**. Panels **(a, c)** show simulations varying $c$ at fixed $h = 1$. The larger the $c$, the more intermittent the behaviour of the fast scales. Panels **(b, d)** show simulations varying the coupling $h$ for fixed $c = 10$. When $h = 0$, there is no small-scale dynamics. $y$ axes are in arbitrary units, and time series are shifted for better visibility.

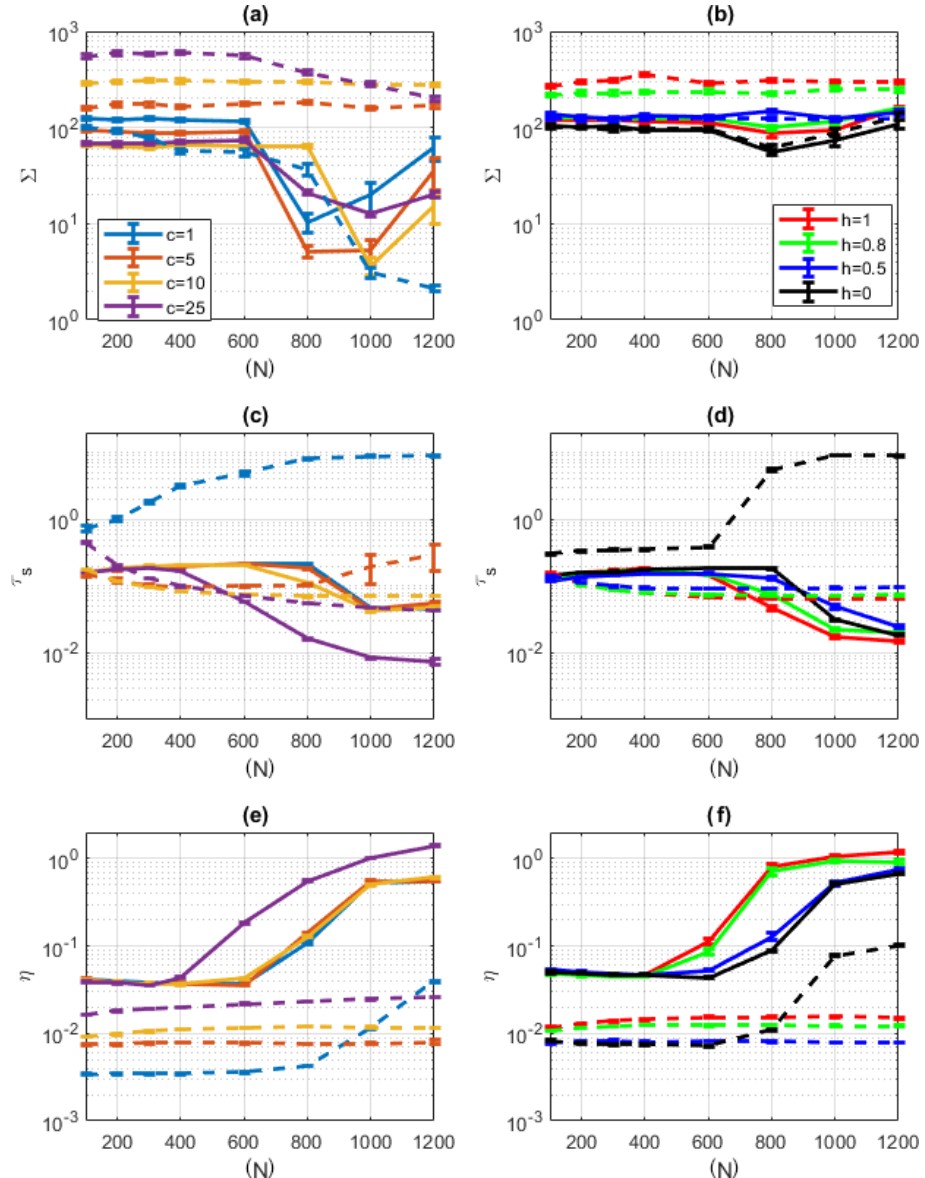

**Figure 7.** Lorenz 1996 ESN prediction performance for $u(t) \equiv \sum_{i=1}^{I} X_i(t)$. **(a, b)** $\chi^2$ distance $\Sigma$; **(c, d)** the predictability horizon $\tau_s$ with $s = 1$. **(e, f)** The initial error $\eta$ in hPa. In **(a, c, e)** ESN predictions are made varying $c$ at fixed $h = 1$. In **(b, d, f)** ESN predictions are made varying $h$ at fixed $c = 10$. Continuous lines show ESN prediction performance made considering $X$ variables only, dotted lines considering both $X$ and $Y$ variables.

orthogonal functions (EOFs) of SLP and a wealth of other atmospheric features (Hurrell, 1995; Moore et al., 2013), ranging from teleconnection patterns to storm track activity to atmospheric blocking, and can be diagnosed from the SLP field.

⁵ The training dataset consists therefore of a gridded time series SLP(t) consisting of ∼ 33 000 TS2 time realizations of the pressure field over a grid of spatial size 72 longitudes × 73 latitudes. Our observable $u(t) \equiv \langle \text{SLP}(t) \rangle_{\text{lon,lat}}$ where brackets indicate the spatial average. In addition to ¹⁰ the time moving-average filter, we also investigate the ef-

fect of spatial coarse graining of the SLP fields by a factor $c$ and perform the learning on the reduced fields. We use the nearest neighbour approximation, which consists in taking from the original dataset the closest value to the coarse grid. ¹⁵ Compared with methods based on averaging or dimension reduction techniques such as EOFs, the nearest neighbour approach has the advantage of not removing the extremes (except if the extreme is not in one of the closest grid points) and preserve cyclonic and anticyclonic structures. For $c = 2$ ²⁰ we obtain a horizontal resolution of 5° and for $c = 4$ a resolution of 10°. For $c = 4$ the information on the SLP field

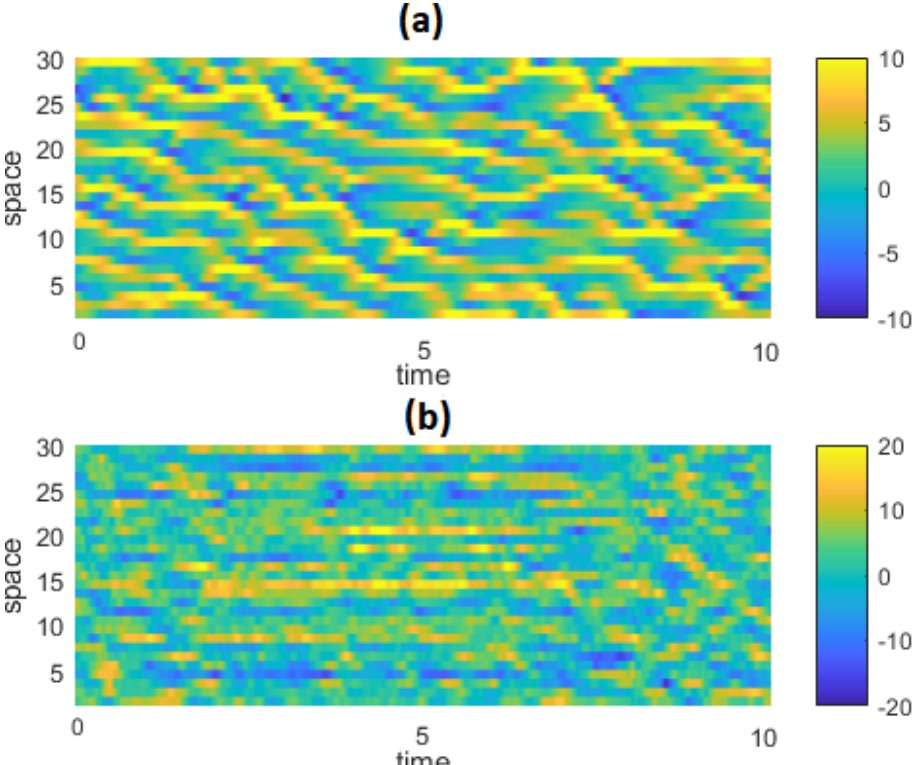

**Figure 8.** Example of **(a)** target Lorenz 1996 spatio-temporal evolution of large-scale variables $X$ for $c = 1, h = 1$ and **(b)** ESN prediction realized with $N = 800$ neurons. Note that the colours are not on the same scale for the two panels.

close to the poles is lost. However, in the remainder of the geographical domain, the coarse-grained fields still capture the positions of cyclonic and anticyclonic structures. Indeed, as shown in Faranda et al. (2017), this coarse-grained field still preserves the dynamical properties of the original one. There is therefore a certain amount of redundant information on the original 2.5° horizontal-resolution SLP fields.

The dependence of the quality of the prediction for the sea-level pressure NCEPv2 data on the coarse-graining factor $c$ and on the moving-average window size $w$ is shown in Fig. 9. We show the results obtained using the residuals (Eq. 10) as the exact method is not straightforwardly adaptable to systems with both spatial and temporal components. Figure 9a–c show the distance from the invariant density using the $\chi^2$ distance $\Sigma$. Here it is clear that by increasing $w$, we get better forecasts with smaller network sizes $N$. A large difference for the predictability expressed as predictability horizon $\tau_s$, $s = 1.5$ hPa (Fig. 9d–f) emerges when SLP fields are coarse-grained. We gain up to 10 h in the predictability horizon with respect to the forecasts performed on the original fields ($c = 0$). This gain is also reflected by the initial error $\eta$ (Fig. 9g–i). Note also that $\Sigma$ blow-up for larger $N$ is related to the long-time instability of the ESN. The blow only affects global indicator $\Sigma$ and not $\tau_s$ and $\eta$, which are computed as short-term properties. From the combination of all the indicators, after a visual inspection, we can identify the

best set of parameters: $w = 12$ h, $N = 200$ and $c = 4$. Indeed, this is the case such that with the smallest network we get almost the minimal $\chi^2$ distance $T$, the highest predictability (32 h) and one of the lowest initial errors. We also note that, for $c = 0$ (panels c and i), the fit always diverges for small network sizes.

We compare in detail the results obtained for two 10-year predictions with $w = 0$ h and $w = 12$ h at $N = 200$ and $c = 4$ fixed. At the beginning of the forecast time (Supplement Video 1), the target field (panel a) is close to both that obtained with $w = 0$ h (panel b) and $w = 12$ h (panel c). When looking at a very late time (Supplement Video 2), of course we do not expect to see agreement among the three datasets. Indeed, we are well beyond the predictability horizon. However, we note that the dynamics for the run with $w = 0$ h is steady: positions of cyclones and anticyclones barely evolve with time. Instead, the run with $w = 12$ h shows a richer dynamical evolution with generation and annihilation of cyclones. A similar effect can be observed in the ESN prediction of the Lorenz 1996 system shown in Fig. 8b where the quasi-horizontal patterns indicate less spatial mobility than the original system (Fig. 8a).

In order to assess the performances of the two ESNs with and without moving averages in a more quantitative way, we present the probability density functions for $u(t) \equiv \langle \text{SLP}(t) \rangle_{\text{lon,lat}}$ in Fig. 10a. The distribution obtained for the

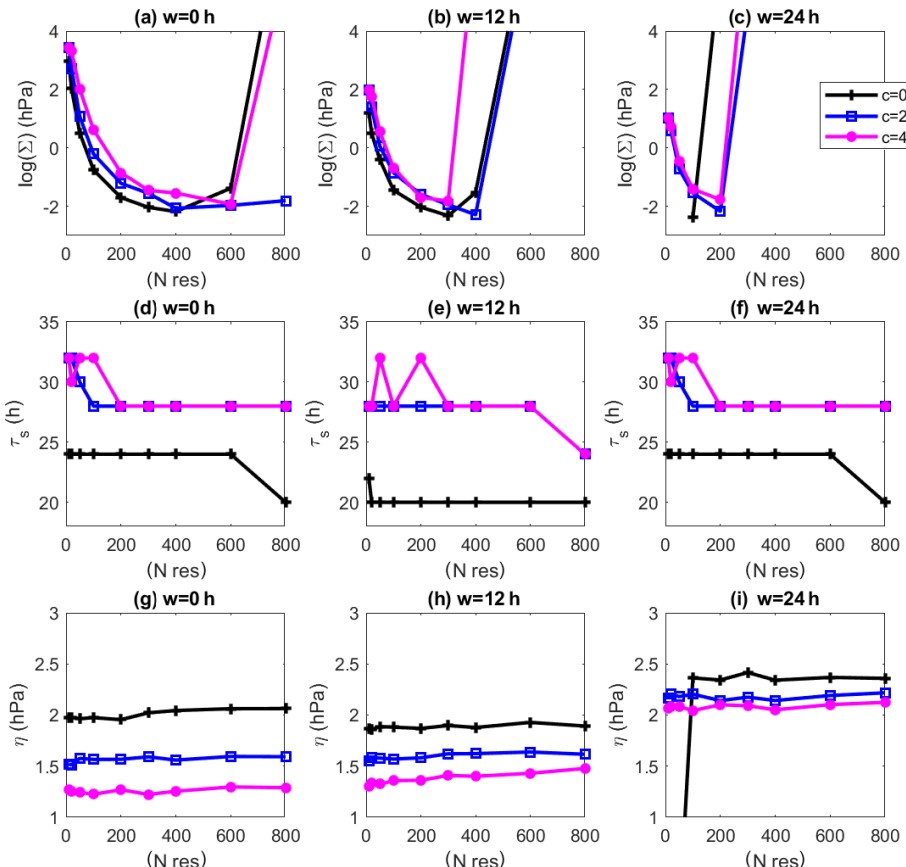

**Figure 9.** Dependence of the quality of the results for the prediction of the sea-level pressure NCEPv2 data on the coarse-graining factor $c$ and on the moving-average window size $w$. The observable used is $u(t) \equiv \langle \mathrm{SLP}(t) \rangle_{\mathrm{lon,lat}}$. **(a–c)** $\chi^2$ distance $\log(\Sigma)$; **(d–f)** predictability horizon (in hours) $\tau_s$, $s = 1.5\,\mathrm{hPa}$; **(g–i)** logarithm of initial error $\eta$. Different coarse-graining factors $c$ are shown with different colours. **(a, d, g)** $w = 0$, **(b, e, h)** $w = 12\,\mathrm{h}$, **(c, f, i)** $w = 24\,\mathrm{h}$.

moving average $w = 12\,\mathrm{h}$ matches better than the run $w = 0\,\mathrm{h}$ that of the target data. Figure 10b–d show the Fourier power spectra for the target data, with the typical decay of a turbulent climate signal. The non-filtered ESN simulation $W = 0$ shows a spectrum with very low energy for high frequency and an absence of the daily cycle (no peak at value $10^0$). The simulation with $w = 12\,\mathrm{h}$ also shows a lower energy for weekly or monthly timescales, but it is the correct peak for the daily cycle and the right energy at subdaily timescales. Therefore, the spectral analysis also shows a real improvement in using moving-average data.

## 4   Discussion

We have analysed the performances of ESNs in reproducing both the short- and long-term dynamics of observables of geophysical flows. The motivation for this study came from the evidence that a straightforward application of ESNs to high-dimensional geophysical data (such as the 6-hourly global gridded sea-level pressure data) does not yield the same result quality obtained by Pathak et al. (2018) for the

Lorenz 1963 and Kuramoto–Sivashinsky models. Here we have investigated the causes of this behaviour and identified two main bottlenecks: (i) intermittency and (ii) the presence of multiple dynamical scales, which both appear in geophysical data. In order to illustrate this effect, we have first analysed two low-dimensional systems, namely the Lorenz (1963) and Manneville (1980) equation. To mimic multiple dynamical scales, we have added noise terms to the dynamics. The performances of ESNs in predicting rapidly drop when the systems are perturbed with noise. Filtering the noise allows us to partially recover predictability. It also enables us to simulate some qualitative intermittent behaviour in the Pomeau–Manneville dynamics. This feature could be explored by changing the degree of intermittency in the Pomeau–Manneville map as well as performing parameter tuning in ESNs. This is left for future work. Our study also suggests that deterministic ESNs with a smooth, continuous activation function cannot be expected to produce trajectories that look spiking/stochastic/rapidly changing. Most previous studies on ESNs (e.g. Pathak et al., 2018) were handling relatively smooth signals and not such rapidly changing

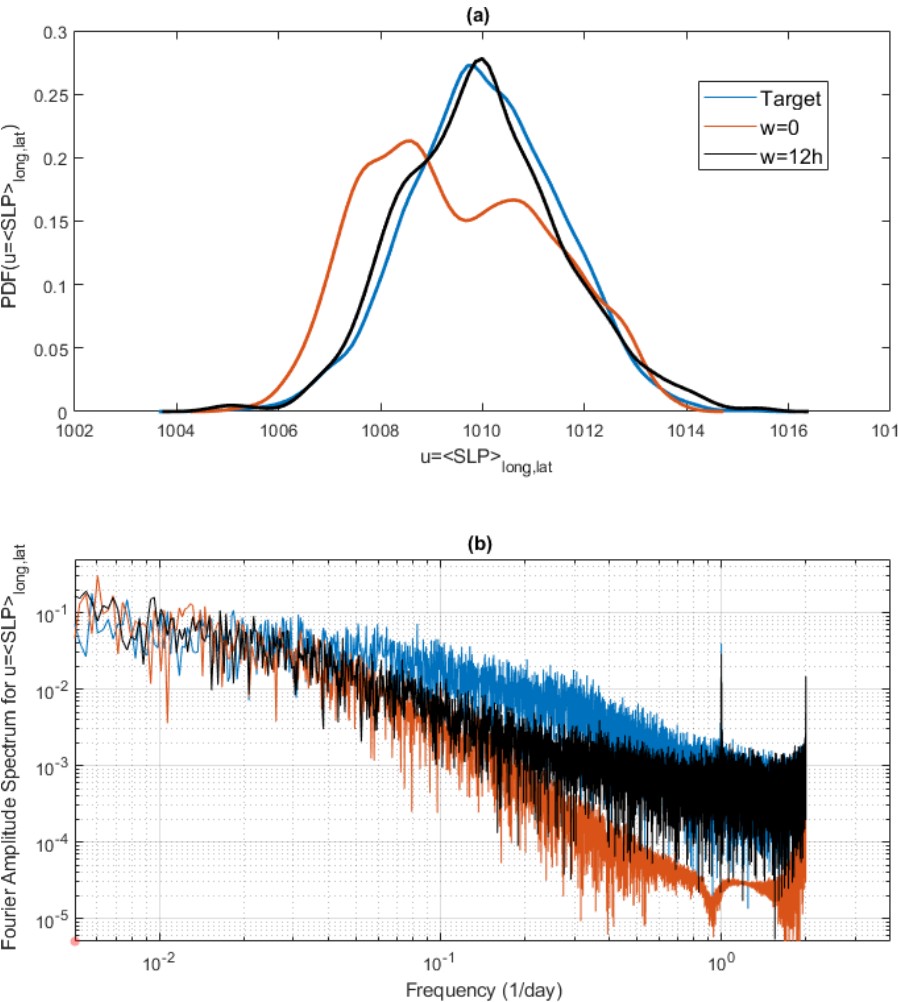

**Figure 10. (a)** Probability density function and **(b)** Fourier power spectra for $u(t) \equiv \langle SLP(t) \rangle_{lon,lat}$ for the target NCEPv2 SLP data (blue), an ESN with $c = 4$ and $w = 0\,h$ (red), and an ESN with $c = 4$ and $w = 12\,h$ (black).

signals. Although it does not come as a surprise that utilizing the ESN on the time-averaged dynamics and then adding a stochastic residual improve performance, the main insights are the intricate dependence of the ESN performance on the noise structure and the fact that, even for a non-smooth signal, ESNs with hyperbolic tangent functions can be used to study systems that have intermittent or multiscale dynamics. Here we have used a simple moving-average filter and shown that a careful choice of the moving-average window can enhance predictability. As an intermediate step between the low-dimensional models and the application to the sea-level pressure data, we have analysed the ESN performances on the Lorenz (1996) system. This system was introduced to mimic the behaviour of the atmospheric jet at mid latitudes and features a lattice of large-scale variables, each connected to small-scale variables. Both the coupling between large and small scales and intermittency can be tuned in the model, giving rise to a plethora of behaviours. For the Lorenz 1996 model, we did not have to apply a moving-average filter to

the data, as we can train the ESN on the large-scale variables only. Our computations have shown that, whenever the small scales are intermittent or the coupling is strong, learning the dynamics of the coarse-grained variable is more effective, both in terms of computation time and performances. The results also apply to geophysical datasets: here we analysed the atmospheric circulation, represented by sea-level pressure fields. Again we have shown that both a spatial coarse graining and a time moving-average filter improve the ESN performances.

Our results may appear rather counterintuitive, as the weather and climate modelling communities are moving towards extending simulations of physical processes to small scales. As an example, we cite the use of highly resolved convection-permitting simulations (Fosser et al., 2015) as well as the use of stochastic (and therefore non-smooth) parameterizations in weather models (Weisheimer et al., 2014). We have, however, a few heuristic arguments for why the coarse-gaining and filtering operations should improve

the ESN performances. First, the moving-average operation helps both in smoothing the signal and providing the ESN with wider temporal information. In some sense, this is reminiscent of the embedding procedure (Cao, 1997), where the signal behaviour is reconstructed by providing not only information on the previous time step, but also on previous times depending on the complexity. The filtering procedure can also be motivated by the fact that the active degrees of freedom for the sea-level pressure data are limited. This has been confirmed by Faranda et al. (2017) by coarse graining these data and showing that the active degrees of freedom are independent of the resolution in the same range explored in this study. Therefore, including small scales in the learning of sea-level pressure data does not provide additional information on the dynamics and push towards overfitting and saturating the ESN with redundant information. The latter consideration also places some caveats on the generality of our results: we believe that this procedure is not beneficial whenever a clear separation of scales is not achievable, e.g. in non-confined 3D turbulence. Moreover, in this study, three sources of stochasticity were present: (i) in the random matrices and reservoir, (ii) in the perturbed initial conditions and (iii) in the ESN simulations when using moving-average-filtered data with sampled $\delta \boldsymbol{x}$ components. The first one is inherent to the model definition. The perturbations of the starting conditions allow characterization of the sensitivity of our ESN approach to the initial conditions. The stochasticity induced by the additive noise $\delta \boldsymbol{x}$ provides a distributional forecast at each time $t$. Although this latter noise can be useful for simulating multiple trajectories and evaluating their long-term behaviour, in practice, i.e. in the case where an ESN would be used operationally to generate forecasts, one might not want to employ a stochastic formulation with an additive noise but rather the explicit and deterministic formulation in Eq. (12). This exemplifies the interest of our ESN approach for possible distinction between forecasts and long-term simulations and therefore makes it flexible to adapt to the case of interest.

Our results, obtained on ESNs, should also be distinguished from those obtained using other RNN approaches. Vlachas et al. (2020) and Levine et al. (2021) TS3 suggest that, although ESNs have the capability for memory, they often struggle to represent it when compared to fully trained RNNs. This essentially defeats the purpose of ESNs, as they are supposed to *learn* memory. In particular, it will be interesting to test whether all experiments reported here could be repeated with a simple artificial neural network, Gaussian process regression, random feature map, or other data-driven function approximator that does not have the dynamical structure of RNNs/ESNs (Cheng et al., 2008; Gottwald and Reich, 2021). In future work, it will be interesting to use other learning architectures and other methods of separating large- from small-scale components (Wold et al., 1987; Froyland et al., 2014; Kwasniok, 1996). Finally, our results give a more formal framework for applications of machine learning techniques on geophysical data. Deep-learning approaches have proven useful in performing learning at different timescales and spatial scales whenever each layer is specialized in learning some specific features of the dynamics (Bolton and Zanna, 2019; Gentine et al., 2018). Indeed, several difficulties encountered in the application of machine learning to climate data could be overcome if the appropriate framework is used, but this requires a critical understanding of the limitations of the learning techniques.

## Appendix A: Numerical code

We report here the MATLAB code used for the computation of the echo state network. This code is adapted from the original code available here: https://mantas.info/code/
simple_esn/ (last access: 28 August 2021).

## A1   ESN training

```
function [Win, W, Wout]=ESN_training(data,Nres)

%This function train the Echo State network using the data provided.
%INPUTS:
%data: a matrix of the input data to train, arranged as space X time

%Nres: the number of neurons N to be used in the training
%OUTPUTS:
%Win: the input weight matrix which consists of random weights

%W: the network of neurons
20 %Wout: the output weights, they are adjusted to match the next iterations

inSize = size(data,1);
trainLen= size(data,2);
Win = (rand(Nres,1+inSize)-0.5) .* 1;
W = rand(Nres,Nres)-0.5;
% normalizing and setting spectral radius

opt.disp = 0;
rhoW = abs(eigs(W,1,'LM',opt));
W = W .* ( 1.25 /rhoW);
% memory allocation
X = zeros(1+inSize+Nres,trainLen-1);
Yt = data(:,2:end)';
x = zeros(Nres,1);
for t = 1:trainLen-1
u = data(:,t);
x = tanh( Win*[1;u] + W*x );
X(:,t) = [1;u;x];
end
reg = 1e-8; % regularization coefficient
Wout = ((X*X' + reg*eye(1+inSize+Nres)) \ (X*Yt))';

end
```

## A2   ESN prediction

```
function [Y_pred]=ESN_prediction(data,Win, W, Wout)

% This function returns the recurrent Echo State Network prediction

%INPUT:
%data: the full data matrix of the data to predict in the form (space*time)

%Win: input weights
%W: neurons matrix
%Wout: output weights
%OUTPUT:
%Y_pred: the ESN prediction
Y_pred = zeros(size(data,1),size(data,2) );

x = zeros(size(W,1),1);
u=data(:,1);
for t = 1:size(data,2)
x = tanh( Win*[1;u] + W*x );
y = Wout*[1;u;x];
Y_pred(:,t) = y;
u = y;
end
end
```

*Code and data availability.* The numerical code used in this article is provided in Appendix A (https://mantas.info/code/simple_esn/, last access: 28 August 2021, Lukoševičius, 2021).TS4

*Video supplement.* .TS5

*Author contributions.* DF, MV, PY, ST and VG conceived this study. DF, AH, CNL and GC performed the analysis. FMEP designed and performed the statistical tests. All the authors contributed to writing and discussing the results of the paper.

*Competing interests.* The authors declare that they have no conflict of interest.

*Acknowledgements.* We thank Barbara D'Alena, Julien Brajard, Venkatramani Balaji, Berengere Dubrulle, Robert Vautard, Nikki Vercauteren, Francois Daviaud, and Yuzuru Sato for useful discussions. We also thank Matthew Levine and three anonymous referees for useful suggestions on the manuscript. This work is supported by CNRS INSU-LEFE-MANU grant "DINCLIC".

*Financial support.* This research has been supported by the CNRS-INSU (LEFE-MANU, grant no. DINCLIC).

*Review statement.* This paper was edited by Amit Apte and reviewed by Matthew Levine and three anonymous referees.

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

**Remarks from the typesetter**

TS1    Please give an explanation of why this needs to be changed. We have to ask the handling editor for approval. Thanks.

TS2    Please give an explanation of why this needs to be changed. We have to ask the handling editor for approval. Thanks.

TS3    Please confirm reference.

TS4    Please confirm section.

TS5    Is it possible to provide DOIs for the video supplements? Please also provide a text for this section. Thank you.

TS6    This reference is not mentioned in the text anymore. Should it be deleted?

TS7    Please confirm all authors' names and initials.

TS8    Please provide location and exact date of the conference.

TS9    Please provide place of publication and more information if possible.

TS10    Please confirm reference list entry that substitutes Basu et al., 2019.

TS11    Please confirm reference.

TS12    Please check DOI.

TS13    Please confirm authors names and initials.

TS14    Please confirm reference list entry.

TS15    Please confirm reference.

TS16    Please confirm reference list entry for the code in the code availability section.

TS17    Please confirm publisher.

TS18    Please check; according to https://images.webofknowledge.com/WOK46P9/help/WOS/P_abrvjt.html, this is the correct abbreviation.

TS19    Please confirm DOI.

TS20    Please confirm the date.

TS21    Please confirm the date.