# Peer review of "Enhancing geophysical flow machine learning performance via scale separation"

_Nonlinear Processes in Geophysics, 2020_

## Referee Comment (RC1) · Anonymous Referee #1 · 14 Oct 2020

**1   General Comments**

This manuscript explores the effectiveness of echo-state-networks for a hierarchy of problems. It explores 3 "toy" dynamical systems and then applies the methodology to a data driven weather prediction task. They evaluate both the equilibrium distribution (using a Xi-squared analysis) and initial value forecasts using root-mean-squared error based metrics. By these metrics, they claim that filtering the data before training an ESN generally improves these metrics in cases where the underlying dynamics are "intermittent" or show strong "coupling between timescales". For all the problems except for Lorenz 96 (L96), they pre-filter with moving averages, whereas for L96 they take advantage of the built-in scale separation between the large-scale and small-scale

variables.

Overall, I thought the results were interesting and relevant to geophysical problems which often feature intermittent and multiscale dynamics, but was not convinced that their claims were valid. See my comments below.

1. The quality of presentation should be improved

    (a) In a few cases, the color schemes used were not intelligible to colorblind readers, which significantly hampered my ability to understand their results. There are many multi-panel figures, which are explained only briefly in the text.
    (b) Notation is used inconsistently and unclearly in some places. Also, this paper introduces redundant notation. Vector and scalar quantities are not differentiated clearly.
    (c) The literature review in the introduction was incomplete in a few places. Also, for an article in a geophysical science journal, concepts like CNNs, RNNs, ESNs should all be clearly defined and differentiated from one another. The introduction sometimes incorrectly conflates these concepts.
    (d) The conclusion contains many helpful motivations that could have helped guide me through the introduction and the methods sections.

2. Their ESNs appear to fail to meaningfully reproduce the time series of the Pomeau-Manneville (fig 5) or Lorenz 96 (fig 8) examples. As with any negative result, it is unclear whether some minor methodological improvement could fix it, so I am not sure what insights these examples provides. In particular, some authors have demonstrated substantially nearly optimal performance with data-driven techniques for Lorenz 96 (Gagne, et. al. 2020) and with ESNs for similar Kuramoto-Shivashinksy model (Pathak, et. al 2018). Were the authors able to replicate the success of these previous studies?
3. The Xi-squared testing procedure seems suspect. It mixes a parametric test (Xi-squared) with a boot-strapping based test. Is there any support for this technique in the literature? It would be preferable to use a more well-known statistical test for this problem (e.g. Wilcoxon Rank sum, Kolmogorov-Smirnov).

4. The sea-level pressure example was compelling.

**2 Specific Comments**

Title: "Boosting performance"

This is a quibble, but "boosting" has a rather specific meaning in the machine learning literature https://en.wikipedia.org/wiki/Boosting_(machine_learning). This could be misleading.

L8: "with an optimal choice of spatial coarse grain and time filtering"

With an optimal choise of spatial coarse-graining

L20. Buchanan.

How does this PhD dissertation relate to the previous assertion. Please be more specific.

L26. Gentine

There are many other articles on parameterizations which should be mentioned e.g. (Brenowitz and Bretherton 2018, 2019; Yuval and O'Gorman 2020; Kransopalky 2005, 2013; Gettleman et. al 2020).

L27. This introduction should also mention (Rasp et. al 2020; Weyn et.al 2019) for the pure weather prediction problem

L40. "Recent examples include... convolutional neural ntworks, ..."

With the previous sentence in mind, this wording implies that convolutional neural networks are a type of RNN. I believe the references all used feed-forward architectures.

L65. "Previous results (Scher, 2018; Dueben and Bauer, 2018; Scher and Messori, 2019) suggest that RNN simulations"

Again, I don't think these papers all studied RNNs. At least some used feed-forward architectures.

L73-90. Overall, this description does not clarify what ESNs are, and why they work outperform traditional RNNs for some problems (e.g. the vanishing gradients problem).

L90: "We estimate w_out via a ridge regression with lambda="

How was this parameter chosen? ESNs are very sensitive to this parameters, and the optimal parameter may vary from problem to problem. This could potentially explain the poor performance on the L96 and Pomeau-Manneville examples below.

L98. "Let, U be ..."

For readibility, try to re-use previously introduced notation to avoid introducing too many new symbols. For instance "v" is the same as "r" in eq 1-4.

Are theses tests univariate? The equations are multivariate.

L120: "we observed excessive rejection rates"

How do you quantify this?

L121: "we use 10000 samples"

What is "a sample". Is it a single time step of r(t) above (e.g. a K-dimensional vector)? Is it the number of timesteps or is it the number of timesteps times K? This would be clearer if described in terms of the notation used in Eqs 1-4.

L135: This formula seems odd. I would normally define predictability by computing RMSE versus the truth for a single timestep. In this case they compute the average

MSE accumulated over several timesteps. Also, this formula only makes sense for scalar u and v, but I thought we are in the vector setting?

Section 2.2: It is unclear why this moving average is described here. It would be clearer if the introduction had introduced a broad outline of the paper.

248: "Performances are again better when using the exact formula (Figure 4b,e,h) than using the residuals $\delta$u (Figure 4c,f,i)."

It would be helpful to refer to Eq 11 here.

Line 250: "ESN simulations do not reproduce the intermittency in the average of the target signal. They only show some second order intermittency in the fluctuations."

Is "the average" supposed to mean "the moving average" rather than "time average"? Is "second order intermittency?". Is this a formal concept?

L270. Forward Euler time steppers are notoriously inaccurate. Why not use a more advanced time stepper (e.g. Runge Kutta) for better accuracy? There are many convenient software packages for integrating ODEs with better schemes (e.g. ode45 in MatLab).

What is N? It must be network size, but given all the notational changes it is hard to be sure.

Line331: "We show the results using the residuals (Eq. 9)"

Why not show the results with the "exact method" (Eq. 11)? It seems the earlier results implied this technique was more effective.

Figure 10 b-d. These panels all look different. I don't see much reason to prefer panel d to c. Could the authors present a more convincing visualization for the claimed improvement of the moving average filter? Maybe a single power-spectra plot would be more succinct, especially since the author's don't comment on the timing the high-frequency vs low-frequency results.

Line 373. "For the Lorenz 1996 mode, we did not apply a moving average filter to the data,..."

It would have been nice to see this motivation described in Section 3.

**3  Technical corrections**

L73. 'Reservoir compution"'

There is a missing quote.

L74. "The principle of Reservoir computing"

Does "Reservoir" need to be capitalized here? If so, I would expect "computing" to be capitalized as well. "reservoir" is not always capitalized in this manuscript.

L76. "In our study *ESNs* are implemented"

L77. "The code is given *in the* appendix

L97: "to this purpose" –> "for this purpose"

L239: "we find the best match... are obtained for w=3"

Correct "are" to "is".

249: "Figure 5a)"

Remove the parenthesis

Line275. "Figure 6.b,d)"

This should read "Figure 6 b,d". Figures should be referred to with a consistent convention.

L288. "distance T".

Do the authors mean $\Sigma$? T is the length of the time series.

Figure 8: The text in this graphic is fuzzy. Please save at a higher resolution.

Figure 2a: This plot has too many curves. Red-green is bad for colorblind readers. It is hard to see the author's point.

Figure 3, 4: These colorscales are not legible for colorblind readers. I could not interpret these figures and relied on the author's textual description of the results. I suggesting using "viridis" or another sequential colorbar.

**4   References**

Pathak, J., Hunt, B., Girvan, M., Lu, Z., & Ott, E. (2018). Model-Free Prediction of Large Spatiotemporally Chaotic Systems from Data: A Reservoir Computing Approach. Physical Review Letters, 120(2). https://doi.org/10.1103/physrevlett.120.024102

Gagne, D. J., II, Christensen, H. M., Subramanian, A. C., & Monahan, A. H. (2020). Machine Learning for Stochastic Parameterization: Generative Adversarial Networks in the Lorenz '96 Model. Journal of Advances in Modeling Earth Systems, 12(3). https://doi.org/10.1029/2019ms001896

Yuval, J. & O'Gorman, P. A. Stable machine-learning parameterization of subgrid processes for climate modeling at a range of resolutions. *Nat. Commun.* **11**, 3295 (2020)

Brenowitz, N. D. & Bretherton, C. S. Spatially Extended Tests of a Neural Network Parametrization Trained by Coarse‐graining. *J. Adv. Model. Earth Syst.* (2019) doi:10.1029/2019MS001711

Krasnopolsky, V. M., Fox-Rabinovitz, M. S. & Chalikov, D. V. New Approach to Calculation of Atmospheric Model Physics: Accurate and Fast Neural Network Emulation of Longwave Radiation in a Climate Model. *Mon. Weather Rev.* **133**, 1370–1383 (2005)

[Figure]

Krasnopolsky, V. M., Fox-Rabinovitz, M. S. & Belochitski, A. A. Using Ensemble of Neural Networks to Learn Stochastic Convection Parameterizations for Climate and Numerical Weather Prediction Models from Data Simulated by a Cloud Resolving Model. *Advances in Artificial Neural Systems* **2013**, e485913 (2013)

Brenowitz, N. D. & Bretherton, C. S. Prognostic Validation of a Neural Network Unified Physics Parameterization. *Geophys. Res. Lett.* **17**, 2493 (2018)

O'Gorman, P. A. & Dwyer, J. G. Using Machine Learning to Parameterize Moist Convection: Potential for Modeling of Climate, Climate Change, and Extreme Events. *J. Adv. Model. Earth Syst.* **10**, 2548–2563 (2018)

Gettleman et. al. Machine Learning the Warm Rain Process.

Rasp, S. *et al.* WeatherBench: A benchmark dataset for data‐driven weather forecasting. *J. Adv. Model. Earth Syst.* (2020) doi:10.1029/2020MS002203

Weyn, J. A., Durran, D. R. & Caruana, R. Can Machines Learn to Predict Weather? Using Deep Learning to Predict Gridded 500‐hPa Geopotential Height From Historical Weather Data. *J. Adv. Model. Earth Syst.* **11**, 2680–2693 (2019)

---

## Referee Comment (RC2) · Anonymous Referee #2 · 4 Nov 2020

**Review - "Boosting performance in machine learning of turbulent and geophysical flows via scale separation"**

November 4, 2020

The authors are utilizing Echo State Networks to predict filtered dynamics in the perturbed Lorenz 1963 equations, the Pomeau-Manneville 89 intermittent map, and the Lorenz 1996 equations. A moving average filter is utilized for scale separation in time. The filtered dynamics are smoother and easier to predict. A residual term is added, either sampled from the training data, or based on an analytic formula derived from the moving average filter. Assuming that the filter width is smaller than the associated large timescales of the processes involved, the large scale processes can be successfully predicted. The authors claim that modeling only the spatially coarse grained and time averaged state can boost performance of ESN. However, the generalization of this argument to more realistic systems is not sufficiently supported by the results, as elaborated in the comment section below.

The idea of utilizing a moving average filter for noise reduction and scale separation, or spatial coarse graining is known. I am not sure that the novelty of the paper to apply ESNs to (spatially/time) filtered dynamics, is enough to guarantee publication in the journal. The effect of the unmodeled dynamics (the information lost during filtering) is not taken into account in the model. In most interesting applications, the effect of the unmodeled modes is the problem, and a field of study by itself (closure models in turbulence, small scale models in weather etc.).

**1 Comments**

1. In the three-dimensional Lorenz system, it is logical that the moving average filter produces better results. By construction, noise is added to the system. It does not come as a surprise that the ESN predicting the filtered dynamics (which are smoother) and augmented with the random residual terms, shows superior performance. However, there is no complex multiscale effect taking place, as the whole state information is given to the system (no hidden state, at least nothing is mentioned in the text about it). Moreover, as a reference time-scale, the Lyapunov time of the deterministic system is used, although the system is augmented with noise, which means that the effective Lyapunov time is in essence much shorter, as stochasticity accelerates the divergence of nearby trajectories. In any case, it is important to be critical about the conclusions drawn from this case.

2. In the Pomeau-Manneville intermittent map, it is not a surprise that the ESN cannot capture the dynamics, as they are changing very rapidly, even visually they look completely stochastic. A deterministic ESN with tanh (smooth, continuous) activation function cannot be expected to produce trajectories that look spiking/stochastic/rapidly changing. Most previous studies on ESNs were handling relatively smooth signals, and not such rapidly changing signals. At least the nature of the signal has to be taken into account in the selection of the activation function of the reservoir. Thus, it does not come as a surprise that utilizing the ESN on the time averaged dynamics and then adding a stochastic residual improves performance. As expected, the plain ESN diverges, as demonstrated also in previous studies with such non-smooth signals.

3. In the Lorenz 96 system, as demonstrated in Figure 8, the method fails to capture the long-term climate, as the dynamics predicted by the ESN are clearly different from the groundtruth.

4. In the sea-level pressure, the moving average filter ESN does not achieve any significant improvement based on the results in Figure 9.

5. In the abstract, the authors claim that "multiscale dynamics and intermittency introduce severe limitations on the applicability of recurrent neural networks, both for short-term forecasts, as well as for the reconstruction of the underlying attractor". This is shown for Echo State Networks in the document, but not in general for Recurrent Neural Networks. The argument has to be relaxed to take into account only ESNs, or a relevant reference for other RNN architectures should be given.

6. There is a contradiction in the text, in page 3, the authors state that "We aim at understanding this sensitivity in a deeper way, while assessing the possibility to reduce its impact on prediction through simple noise reduction methods", although one sentence before, they claim that they choose the ESN framework for "...its ability to forecast chaotic time series and its stability to noise". These sentences are contradicting each other. Later in the text, the authors state "Since Echo State Networks are known to be sensitive to noise (see e.g. [34]), ...".

7. The analysis of the performance of the proposed method based on different parameters e.g. intermittency of dynamics/degree of coarse graining, etc. is interesting. However, this is not adequate to warrant publication.

**2 Proposal**

Reject.

---

## Author Comment (AC1) · 18 Dec 2020

This manuscript explores the effectiveness of echo-state-networks for a hierarchy of problems. It explores 3 "toy" dynamical systems and then applies the methodology to a data driven weather prediction task. They evaluate both the equilibrium distribution (using a Xi-squared analysis) and initial value forecasts using root-mean-squared error based metrics. By these metrics, they claim that filtering the data before training an ESN generally improves these metrics in cases where the underlying dynamics are "intermittent" or show strong "coupling between timescales". For all the problems except for Lorenz 96 (L96), they pre-filter with moving averages, whereas for L96 they take advantage of the built-in scale separation between the large-scale and small-scale variables. Overall, I thought the results were interesting

and relevant to geophysical problems which often feature intermittent and multiscale dynamics, but was not convinced that their claims were valid. See my comments below.

**We thank the reviewer for the appreciation of our work. In the new version of the manuscript we will fully address the recommendations. A detailed answer is provided below.**

MAJOR COMMENTS

1. The quality of presentation should be improved (a) In a few cases, the color schemes used were not intelligible to colorblind readers, which significantly hampered my ability to understand their results. There are many multi-panel figures, which are explained only briefly in the text.

**We will take great care in changing the color scales for colorblind users and we are sorry the reviewer had trouble in understanding some of the results. Multi Panels figures will be discussed in more details in the text.**

(b) Notation is used inconsistently and unclearly in some places. Also, this paper introduces redundant notation. Vector and scalar quantities are not differentiated clearly.

**Following the suggestion of the reviewer (see also answers to technical comments), we will use a more compact and consistent notation.**

(c) The literature review in the introduction was incomplete in a few places. Also, for an article in a geophysical science journal, concepts like CNNs, RNNs, ESNs should

all be clearly defined and differentiated from one another. The introduction sometimes incorrectly conflates these concepts.

**We will take great care to differentiate CNN from RNN and ESN in the new version of the manuscript.**

(d) The conclusion contains many helpful motivations that could have helped guide me through the introduction and the methods sections.

**We will move some of the key concepts presented in the conclusions, in the introduction section.**

2. Their ESNs appear to fail to meaningfully reproduce the time series of the Pomeau-Manneville (fig 5) or Lorenz 96 (fig 8) examples. As with any negative result, it is unclear whether some minor methodological improvement could fix it, so I am not sure what insights these examples provides. In particular, some authors have demonstrated substantially nearly optimal performance with data driven techniques for Lorenz 96 (Gagne, et. al. 2020) and with ESNs for similar Kuramoto-Shivashinksy model (Pathak, et. al 2018). Were the authors able to replicate the success of these previous studies?

**Indeed we are able to reproduce the previous results obtained with ESN for the models outlined by the reviewer. However, in this paper we decide to use the simplest possible ESN (i.e. not the tuned one which indeed provides better performances to the single cases) to perform sensitivity studies on noise level and coarse-graining, in an extended, comprehensive and parameters controlled way. As pointed out by the other referee, a deterministic ESN with smooth, continuous activation function cannot be expected to produce trajectories that look**

**spiking/stochastic/rapidly changing. Most previous studies on ESNs were handling relatively smooth signals, and not such rapidly changing signals. Although it does not come as a surprise that utilizing the ESN on the time averaged dynamics and then adding a stochastic residual improves performance, the main insights is the intricate dependence of the ESN performance on the noise structure and the fact that, even for non-smooth signal, ESN with hyperbolic tanh functions can be used to study systems that have a multiscale dynamics. In the new version of the manuscript we will make these concepts more clear.**

3. The Xi-squared testing procedure seems suspect. It mixes a parametric test (Xisquared) with a boot-strapping based test. Is there any support for this technique in the literature? It would be preferable to use a more well-known statistical test for this problem (e.g. Wilcoxon Rank sum, Kolmogorov-Smirnov).

**We are aware of the limitations of the strategy we decided to adopt, but after considering different strategies, including those suggested by the referee, we decided to still adopt this approach. There are two aspects playing a role in this choice: why to use a chi-squared test and why to conduct a Monte Carlo experiment to determine the critical value. First, we need to clarify that we did not use a bootstrap methodology, which relies on resampling. Instead, we adopt a Monte Carlo simulation approach, so that each of the 10000 samples is generated under the null hypothesis. This is possible because we are considering simulated systems, for which we can obtain as many independent samples as we want. This choice is made necessary by the fact that, due to limits in the length of each time series, we cannot observe the entire invariant distribution of the process, and therefore we cannot use the theoretical distribution of the test statistics under the null hypothesis. This situation would happen in any case, independently on the chosen test statistic, because the problem resides in our inability to observe the invariant distribution. In other words, we run a simulation to obtain tabulated**

values of the distribution of the test statistics under H0 specific to our study, and we would do this for any adopted statistical test. Not using this procedure would make fixing the level of the test not sufficient to control the probability of Type 1 error. The reasons for the choice of the chi-squared test reside mainly in the construction of the test statistics: this is built considering values of the empirical probability density functions (pdf) over the entire domain, i.e. using all the bins of the histograms. Procedures based on ranks such as the Wilcoxon Rank sum, on the other hand, test more specific null hypotheses. The Wilcoxon rank sum tests the null hypothesis of identical distribution against the specific alternative that one of the two distributions exhibits stochastic dominance (this degenerates to a test on the median in case of Gaussian homoskedastic variables); however, this test could end up not rejecting the null hypothesis in case of pdfs with sensibly different shape, but not clear stochastic dominance. In the case of the KS, the test is indeed useful for testing if two distributions are identical in a more general way. However, in our experience the KS test tends to reject the null hypothesis even in presence of substantially trivial differences between the distributions. Other examples of tests that consider the shape of the entire distribution are the Anderson Darling and the Cramer-von-Mises tests. We did see an advantage in using the chi-squared because it is based not on a distance, but on an asymmetric divergence that gives more weight to the reference distribution: since we are not comparing two sample distributions, but a sample distribution and the true distribution of the dynamical system observable, we are willing to place more weight on the latter. These explanations will be added to the new version of the paper.

4. The sea-level pressure example was compelling.

**We thank the referee for the appreciation of the results for the sea-level pressure.**

SPECIFIC COMMENTS

Title: "Boosting performance" This is a quibble, but "boosting" has a rather specific meaning in the machine learning literature https://en.wikipedia.org/wiki/Boosting$_{(machine_learning)}$. $This could be misleading.$

**Thank you for this remark, if possible, we will change the wording "boosting" with "enhancing"**

L8: "with an optimal choice of spatial coarse grain and time filtering" With an optimal choise of spatial coarse-graining

**Corrected**

L20. Buchanan. How does this PhD dissertation relate to the previous assertion. Please be more specific.

**We will give precision about this reference in the new version of the manuscript**

L26. Gentine There are many other articles on parameterizations which should be mentioned e.g. (Brenowitz and Bretherton 2018, 2019; Yuval and O'Gorman 2020; Kransopalky 2005, 2013; Gettleman et. al 2020).

**We will add these references to the new version of the manuscript**

L27. This introduction should also mention (Rasp et. al 2020; Weyn et.al 2019) for the

pure weather prediction problem

**We will add these references to the new version of the manuscript**

L40. "Recent examples include. . . convolutional neural ntworks, . . . " C3 With the previous sentence in mind, this wording implies that convolutional neural networks are a type of RNN. I believe the references all used feed-forward architectures.

**Thank you. This will be corrected in the new version of the manuscript.**

L65. "Previous results (Scher, 2018; Dueben and Bauer, 2018; Scher and Messori, 2019) suggest that RNN simulations" Again, I don't think these papers all studied RNNs. At least some used feed-forward architectures.

**We will clarify this in the new version of the manuscript.**

L73-90. Overall, this description does not clarify what ESNs are, and why they work outperform traditional RNNs for some problems (e.g. the vanishing gradients problem).

**We will clearly state when we talk ESNs or RNN in the new version of the manuscript.**

L90: "We estimate Wout via a ridge regression with lambda=" How was this parameter chosen? ESNs are very sensitive to this parameters, and the optimal parameter may vary from problem to problem. This could potentially explain the poor performance on the L96 and Pomeau-Manneville examples below.

**Ridge minimizes the residual sum of squares plus a shrinkage penalty of lambda multiplied by the sum of squares of the coefficients. As lambda increases, the coefficients approach zero. The coefficients are unregularized when lambda is zero. In the new version of the manuscript we will show that the value of lambda is chosen via cross-validation for the Lorenz 1963. Indeed, we will repeat the procedure to find the good level for the Lorenz 1996 and the PM examples, as the value has been directly taken from the cross validation for the lorenz 1963. Thanks for the suggestion.**

L98. "Let, U be . . . " For readability, try to re-use previously introduced notation to avoid introducing too many new symbols. For instance "v" is the same as "r" in eq 1-4. Are theses tests univariate? The equations are multivariate.

**The tests are all univariate, for the Lorenz 1963 we consider the variable x only. For the Lorenz 1996 we consider one of the variables, since they are all dynamically and statistically equivalent. For the SLP, we consider the spatial average as observables for the test. We will improve the readability as suggested by changing the notation and making clear these points.**

L120: "we observed excessive rejection rates" How do you quantify this?

**We underline that the sentence is actually: "we would observe excessive rejection rates". Here we are underlining that, due to intrinsic limitations, we can construct a chi-squared test, but not use the standard critical values for the distribution of the test statistic, which would produce excessive rejection rates. Therefore, we construct the test statistic in the usual way, but use Monte**

Carlo simulation to obtain the distribution of the test statistic under the null
hypothesis.

L121: "we use 10000 samples" What is "a sample". Is it a single time step of r(t) above
(e.g. a K-dimensional vector)? Is it the number of timesteps or is it the number of
timesteps times K? This would be clearer if described in terms of the notation used in
Eqs 1-4.

**We will specify in the new version of the manuscript that a sample is a series
of a univariate [ as specified in the answer for L98] test observables, and we
consider 10000 samples extracted from the total, longer time series.**

L135: This formula seems odd. I would normally define predictability by computing
RMSE versus the truth for a single timestep. In this case they compute the average
MSE accumulated over several timesteps. Also, this formula only makes sense for
scalar u and v, but I thought we are in the vector setting?

**Root mean squared error is by definition a mean over several values (not neces-
sarily ordered in time): the errors on each single time step are usually averaged
on the whole available sample to compute a single forecasting performance met-
ric. However, if one is interested in assessing a model performance over a given
time horizon on time series data, say $\tau$, the computation of RMSE can be limited
to the period from $t$ to $t + \tau$. For example, suppose we want to evaluate the per-
formance of a statistical model in predicting the next $\tau = 3$ days in a temperature
time series; then, for every day in the sample we would only compute the RMSE
of days t+1, t+2, t+3. Here we do the same, but using several values of $\tau$ to find
the maximum predictability horizon, over which the method loses efficacy.**

Section 2.2: It is unclear why this moving average is described here. It would be clearer if the introduction had introduced a broad outline of the paper.

**We will follow the suggestion of the reviewer.**

L248: "Performances are again better when using the exact formula (Figure 4b,e,h) than using the residuals $\delta u$ (Figure 4c,f,i)." It would be helpful to refer to Eq 11 here.

**Thank you, we will add Eq 11 there.**

L250: "ESN simulations do not reproduce the intermittency in the average of the target signal. They only show some second order intermittency in the fluctuations." Is "the average" supposed to mean "the moving average" rather than "time average"? Is "second order intermittency?". Is this a formal concept?

**What we mean here is that during the intermittent phases, the PM dynamics oscillate in the range (0.2 1) with an average of about 0.6. In the non-intermittent phases, the PM dynamics is stuck near 0. Therefore, the intermittency is on average (shift from 0.6 to 0) and in variance. This will be added to the new version of the manuscript.**

L270. Forward Euler time steppers are notoriously inaccurate. Why not use a more advanced time stepper (e.g. Runge Kutta) for better accuracy? There are many convenient software packages for integrating ODEs with better schemes (e.g. ode45 in MatLab). What is N? It must be network size, but given all the notational changes it is hard to be sure.

**We remind that here the idea is to have exemples close to the atmospheric or climate data: when considering daily or 6 hourly data, as commonly done in**

**climate sciences and analyses, we hardly are in the case of a smooth RK time stepper. We therefore stick to the Euler method for similarity with the actual climate data. This will be added to the text.**

L331: "We show the results using the residuals (Eq. 9)" Why not show the results with the "exact method" (Eq. 11)? It seems the earlier results implied this technique was more effective.

**Unfortunately the "exact method" cannot be used for the SLP NCEP data. Indeed this dynamics has a spatial component that the "exact" method cannot take into account the spatial component. This is now specified in the paper.**

Figure 10 b-d. These panels all look different. I don't see much reason to prefer panel d to c. Could the authors present a more convincing visualization for the claimed improvement of the moving average filter? Maybe a single power-spectra plot would be more succinct, especially since the author's don't comment on the timing of the high-frequency vs low-frequency results.

**The wavelet methodology is a more sophisticated representation of a spectrum. Since the referee demands it, we will simplify this part by replacing wavelet spectra with conventional spectra.**

L373. "For the Lorenz 1996 mode, we did not apply a moving average filter to the data,. . . " It would have been nice to see this motivation described in Section 3.

**We will add this motivation in Section 3.**

TECHNICAL CORRECTIONS

L73. 'Reservoir compution"' There is a missing quote. L74. "The principle of Reservoir computing" Does "Reservoir" need to be capitalized here? If so, I would expect "computing" to be capitalized as well. "reservoir" is not always capitalized in this manuscript. L76. "In our study ESNs are implemented" L77. "The code is given in the appendix L97: "to this purpose" –> "for this purpose" L239: "we find the best match. . . are obtained for w=3" Correct "are" to "is". 249: "Figure 5a)" Remove the parenthesis Line275. "Figure 6.b,d)" This should read "Figure 6 b,d". Figures should be referred to with a consistent convention. L288. "distance T". C6 Do the authors mean $\Sigma$? T is the length of the time series. Figure 8: The text in this graphic is fuzzy. Please save at a higher resolution. Figure 2a: This plot has too many curves. Red-green is bad for colorblind readers. It is hard to see the author's point. Figure 3, 4: These colorscales are not legible for colorblind readers. I could not interpret these figures and relied on the author's textual description of the results. I suggesting using "viridis" or another sequential colorbar.

**Thank you, technical corrections will be implemented. Colorscales replaced as demanded for colorblind readers.**

REFERENCES

Pathak, J., Hunt, B., Girvan, M., Lu, Z., Ott, E. (2018). Model-Free Prediction of Large Spatiotemporally Chaotic Systems from Data: A Reservoir Computing Approach. Physical Review Letters, 120(2). https://doi.org/10.1103/physrevlett.120.024102

Gagne, D. J., II, Christensen, H. M., Subramanian, A. C., Monahan, A. H. (2020). Machine Learning for Stochastic Parameterization: Generative Adversarial Networks

in the Lorenz '96 Model. Journal of Advances in Modeling Earth Systems, 12(3). https://doi.org/10.1029/2019ms001896

Yuval, J. O'Gorman, P. A. Stable machine-learning parameterization of subgrid processes for climate modeling at a range of resolutions. Nat. Commun. 11, 3295 (2020)

Brenowitz, N. D. Bretherton, C. S. Spatially Extended Tests of a Neural Network Parametrization Trained by Coarse' Rgraining. ËĞ J. Adv. Model. Earth Syst. (2019) doi:10.1029/2019MS001711

Krasnopolsky, V. M., Fox-Rabinovitz, M. S. Chalikov, D. V. New Approach to Calculation of Atmospheric Model Physics: Accurate and Fast Neural Network Emulation of Longwave Radiation in a Climate Model. Mon. Weather Rev. 133, 1370–1383 (2005)

Krasnopolsky, V. M., Fox-Rabinovitz, M. S. Belochitski, A. A. Using Ensemble of Neural Networks to Learn Stochastic Convection Parameterizations for Climate and Numerical Weather Prediction Models from Data Simulated by a Cloud Resolving Model. Advances in Artificial Neural Systems 2013, e485913 (2013)

Brenowitz, N. D. Bretherton, C. S. Prognostic Validation of a Neural Network Unified Physics Parameterization. Geophys. Res. Lett. 17, 2493 (2018) O'Gorman, P. A. Dwyer, J. G. Using Machine Learning to Parameterize Moist Convection: Potential for Modeling of Climate, Climate Change, and Extreme Events. J. Adv. Model. Earth Syst. 10, 2548–2563 (2018)

Gettleman et. al. Machine Learning the Warm Rain Process. Rasp, S. et al. Weather-Bench: A benchmark dataset for data' Rdriven weather fore- ËĞ casting. J. Adv. Model. Earth Syst. (2020) doi:10.1029/2020MS002203

Weyn, J. A., Durran, D. R. Caruana, R. Can Machines Learn to Predict Weather? Using Deep Learning to Predict Gridded 500' RhPa Geopotential Height From Historical ËĞ Weather Data. J. Adv. Model. Earth Syst. 11, 2680–2693 (2019)

**We thank the reviewer for the references, which will be properly added to the**

**new version of the manuscript.**

---

## Author Response (AR1)

**Dear Editor,**

We have revised the manuscript by following the suggestion of the reviewers. We provide a new version of our study where figures have been improved, notation has been improved and the additional analyses proposed have been integrated. When changing figures and taking into account comments, we have been in the need to redo more realisations for our experiments. Of course, the new realizations included in some of the figures of this paper show statistically the same results, but they can be slightly visually different then the previous version. Furthermore, we have edited some of the answers initially proposed to the referees to reflect the changes in the manuscript. We believe that when reviewing the new version of the manuscript, the referees should find in the present answer letter a more detailed description of changes made in the manuscript than the answers previously published. We regret that reviewer 2 did not provide useful references or suggestions to improve the quality of the manuscript. A marked-up version of the manuscript showing the implemented changes is attached at the end of this answer letter.

Finally we believe that our editing efforts are reflected in the general improvement of the quality of the paper, which is now more readable and scientifically sound. In particular, new figure 10 shows a clear evidence of the improvement in forecast of atmospheric fields obtained with the filtering procedure.

**Best Regards**

Davide Faranda (on behalf of all the authors)

**RC1** Anonymous Reviewer**

This manuscript explores the effectiveness of echo-state-networks for a hierarchy of problems. It explores 3 "toy" dynamical systems and then applies the methodology to a data driven weather prediction task. They evaluate both the equilibrium distribution (using a Xi-squared analysis) and initial value forecasts using root-mean-squared error based metrics. By these metrics, they claim that filtering the data before training an ESN generally improves these metrics in cases where the underlying dynamics are "intermittent" or show strong "coupling between timescales". For all the problems except for Lorenz 96 (L96), they pre-filter with moving averages, whereas for L96 they take advantage of the built-in scale separation between the large-scale and small-scale variables. Overall, I thought the results were interesting and relevant to geophysical problems which often feature intermittent and multiscale dynamics, but was not convinced that their claims were valid. See my comments below.

**We thank the reviewer for the appreciation of our work. In the new version of the manuscript we have fully addressed the recommendations. A detailed answer is provided below**

**MAJOR COMMENTS**

**1. The quality of presentation should be improved**

(a) In a few cases, the color schemes used were not intelligible to colorblind readers, which significantly hampered my ability to understand their results. There are many multi-panel figures, which are explained only briefly in the text.

We have taken great care in changing the color scales for colorblind users and we are sorry the reviewer had trouble in understanding some of the results. Multi Panels figures have been discussed in more details in the text.

(b) Notation is used inconsistently and unclearly in some places. Also, this paper introduces redundant notation. Vector and scalar quantities are not differentiated clearly.

Following the suggestion of the reviewer (see also answers to technical comments), we have used a more compact and consistent notation. The main confusion probably originated by the misuse of u(t) in the formula of ESN. Indeed for the training we use the vectors x(t) of the dynamical systems, while the observable u(t) is a scalar quantity for a given time. Also in section 2.2 and 2.3 we have replaced u(t) with x(t) because the MA filter is operated on the signal and not on the observable. It is now clearly stated what observable is taken for the computation of the metrics used as diagnostic

(c) The literature review in the introduction was incomplete in a few places. Also, for an article in a geophysical science journal, concepts like CNNs, RNNs, ESNs should all be clearly defined and differentiated from one another. The introduction sometimes incorrectly conflates these concepts.

**We have taken great care to differentiate CNN from RNN and ESN in the new version of the manuscript.**

(d) The conclusion contains many helpful motivations that could have helped guide me through the introduction and the methods sections.

**We have moved some of the key concepts presented in the conclusions, in the introduction section and rephrased a few sentences to make the purpose of the study clearer.**

2. Their ESNs appear to fail to meaningfully reproduce the time series of the Pomeau-Manneville (fig 5) or Lorenz 96 (fig 8) examples. As with any negative result, it is unclear whether some minor methodological improvement could fix it, so I am not sure what insights these examples provides. In particular, some authors have demonstrated substantially nearly optimal performance with data driven techniques for Lorenz 96 (Gagne, et. al. 2020) and with ESNs for similar Kuramoto-Shivashinksy model (Pathak, et. al 2018). Were the authors able to replicate the success of these previous studies?

Indeed we are able to reproduce the previous results obtained with ESN for the models outlined by the reviewer. However, in this paper we decide to use the simplest possible ESN (i.e. not the tuned one which indeed provides better performances to the single cases) to perform sensitivity studies on noise level and coarse-graining, in an extended, comprehensive and parameters controlled way. As pointed out by the other referee, a deterministic ESN with smooth, continuous activation function cannot be expected to produce trajectories that look spiking/stochastic/rapidly changing. Most previous studies on ESNs were handling relatively smooth signals, and not such rapidly changing signals. Although it does not come as a surprise that utilizing the ESN on the time averaged dynamics and then adding a stochastic residual improves performance, the main insights is the intricate dependence of the ESN performance on the noise structure and the fact that, even for non-smooth signal, ESN with hyperbolic tanh functions can be used to study systems that have a multiscale dynamics. In the new version of the manuscript we have added these considerations to the discussion section of the articles.

3. The Xi-squared testing procedure seems suspect. It mixes a parametric test (Xisquared) with a boot-strapping based test. Is there any support for this technique in the literature? It would be preferable to use a more well-known statistical test for this problem (e.g. Wilcoxon Rank sum, Kolmogorov-Smirnov).

We are aware of the limitations of the procedure we decided to adopt, but after considering different strategies, including those suggested by the referee, we decided to still keep this approach. There are two aspects playing a role in this choice: why to use a chi-squared test and why to conduct a Monte Carlo experiment to determine the critical value.

The reasons for the choice of the chi-squared test reside mainly in the construction of the test statistics: this is built to compare the probability distribution, considering differences between the empirical probability density functions (pdf) over the entire domain, in practice the histograms. Under the null hypothesis that the two distributions are the same, their differences in the bins are i.i.d normal random variables, and the chi-squared statistics follows indeed a chi-square distribution with M degrees of freedom, with M the number of bins. Other examples of tests that consider the shape of the entire distribution are the Anderson Darling and the Cramer-von-Mises tests. We did see an advantage in using the chi-squared because it is based not on a distance, but on an asymmetric divergence that gives more weight to the reference distribution: since we are not comparing two sample distributions, but a sample distribution and the true distribution of the dynamical system observable, we are willing to place more weight on the latter. These explanations have been added to the new version of the paper.

Procedures based on ranks such as the Wilcoxon Rank sum, on the other hand, test more specific null hypotheses. The Wilcoxon rank sum tests the null hypothesis of identical distribution against the specific alternative that one of the two distributions exhibits stochastic dominance (this degenerates to a test on the median in case of Gaussian homoskedastic variables); however, this test could end up not rejecting the null hypothesis in case of pdfs with sensibly different shape, but not clear stochastic dominance or shift in median. In the case of the KS, the test is indeed useful for testing if two distributions are identical in a more general way. However, in our experience the KS test tends, at large sample sizes, to over-reject the null hypothesis in presence of very small differences between the distributions.

Concerning the experiment design, we need to clarify that we did not use a bootstrap test, which relies on resampling. Instead, we adopt a Monte Carlo *simulation* approach to approximate the distribution of the test statistics, and then the critical rejection value. Each of the 10000 samples is generated under the null hypothesis, which is possible because we are considering simulated systems, for which we can obtain as many independent samples as we want.

We made this choice because, even simulating relatively long trajectories, we cannot observe the entire invariant distribution of the process, and therefore we cannot use the theoretical chi2(M) distribution of the test statistic. This situation would happen in any case, independently on the chosen test, because the problem resides in our inability to observe the invariant distribution. In other words, we run a simulation to obtain tabulated values of the distribution of the test statistics under H0 specific to our study, and we would do this for any adopted statistical test, as using the exact or asymptotic distribution (depending on the test) would make fixing the level of the test not sufficient to control the probability of Type 1 error.

4. The sea-level pressure example was compelling.

**We thank the referee for the appreciation of the results for the sea-level pressure.**

**SPECIFIC COMMENTS**

Title: "Boosting performance" This is a quibble, but "boosting" has a rather specific meaning in the machine learning literature https://en.wikipedia.org/wiki/Boosting\_(machine\_learning). This could be misleading.

**Thank you for this remark, if possible, we have changed the wording "boosting" with "enhancing"**

L8: "with an optimal choice of spatial coarse grain and time filtering" With an optimal choise of spatial coarse-graining

**Corrected**

L20. Buchanan. How does this PhD dissertation relate to the previous assertion. Please be more specific.

**We have corrected this reference which corresponds to a journal article and not to a Phd dissertation as previously stated.**

L26. Gentine There are many other articles on parameterizations which should be mentioned e.g. (Brenowitz and Bretherton 2018, 2019; Yuval and O'Gorman 2020; Kransopalky 2005, 2013; Gettleman et. al 2020).

**After revising them, we agree that these references contain relevant information for the cited step and we have added them to the new version of the manuscript**

L27. This introduction should also mention (Rasp et. al 2020; Weyn et.al 2019) for the pure weather prediction problem

**We have added these references to the new version of the manuscript**

L40. "Recent examples include... convolutional neural ntworks, ... " C3 With the previous sentence in mind, this wording implies that convolutional neural networks are a type of RNN. I believe the references all used feed-forward architectures.

**Thank you. This have been corrected in the new version of the manuscript**

L65. "Previous results (Scher, 2018; Dueben and Bauer, 2018; Scher and Messori, 2019) suggest that RNN simulations" Again, I don't think these papers all studied RNNs. At least some used feed-forward architectures. We have rephrased this in the new version of the manuscript

L73-90. Overall, this description does not clarify what ESNs are, and why they work outperform traditional RNNs for some problems (e.g. the vanishing gradients problem).

**Again, we have stated clearly the specificity of the ESN at the beginning of the methods section: "In our study we particularly use ESN, a particular case of RNN where the output and the input have the same dynamical**

form. In the ESN approach, neuron layers are replaced by a sparsely connected network, with randomly assigned fixed weights. we harvest reservoir states via a nonlinear transform of the driving input and compute the output weights to create reservoir-to-output connections."

L90: "We estimate w\_out via a ridge regression with lambda=" How was this parameter chosen? ESNs are very sensitive to this parameters, and the optimal parameter may vary from problem to problem. This could potentially explain the poor performance on the L96 and Pomeau-Manneville examples below.

Ridge minimizes the residual sum of squares plus a shrinkage penalty of lambda multiplied by the sum of squares of the coefficients. As lambda increases, the coefficients approach zero. The coefficients are unregularized when lambda is zero. We have tested the dependence on the ridge regression parameter for the Pomeau Manneville map and found that the dependence is small.

The figure above shows the dependence of \phi, log(\tau\_s=1) and log(\eta) from the value of the ridge parameter. No particular dependence is shown, so we stick to the small value \lambda=10^(-8) for the computations shown in this article.

L98. "Let, U be . . . " For readibility, try to re-use previously introduced notation to avoid introducing too many new symbols. For instance "v" is the same as "r" in eq 1-4. Are theses tests univariate? The equations are multivariate.

We believe that the confusion originated by our wrong use of the notation. Now, for each system we have specified the observable used. The tests are all univariate, for the Lorenz 1963 we consider the variable x only. For the Lorenz 1996 we consider one of the variables, since they are all dynamically and statistically equivalent. For the SLP, we consider the spatial average as observables for the test.

L120: "we observed excessive rejection rates" How do you quantify this?

We underline that the sentence is actually: "we would observe excessive rejection rates". Here we are underlining that, due to intrinsic limitations, we can construct a chi-squared test, but not use the standard critical values for the distribution of the test statistic, which would produce excessive rejection rates. Therefore, we construct the test statistic in the usual way, but use Monte Carlo simulation to obtain the distribution of the test statistic under the null hypothesis.

L121: "we use 10000 samples" What is "a sample". Is it a single time step of r(t) above (e.g. a K-dimensional vector)? Is it the number of timesteps or is it the number of timesteps times K? This would be clearer if described in terms of the notation used in Eqs 1-4.

**We have specified in the new version of the manuscript that a sample is a series of a univariate [ as specified in the answer for L98] test observables, and we consider 10000 samples extracted from the total, longer time series.**

L135: This formula seems odd. I would normally define predictability by computing RMSE versus the truth for a single timestep. In this case they compute the average MSE accumulated over several timesteps. Also, this formula only makes sense for scalar u and v, but I thought we are in the vector setting?

**We thank the referee for the comment. Unlike our previous response in the discussion, we acknowledge that Indeed the equation was incorrect, after initially using different error measures, we indeed compute the absolute prediction error (APE), and not the RMSE, for each time step.**

Section 2.2: It is unclear why this moving average is described here. It would be clearer if the introduction had introduced a broad outline of the paper.

**We have followed the suggestion of the reviewer and introduced the moving average in the introduction**

L248: "Performances are again better when using the exact formula (Figure 4b,e,h) than using the residuals  $\delta$  u (Figure 4c,f,i)." It would be helpful to refer to Eq 11 here.

**Thank you, we have added Eq 11 there**

L250: "ESN simulations do not reproduce the intermittency in the average of the target signal. They only show some second order intermittency in the fluctuations." Is "the average" supposed to mean "the moving average" rather than "time average"? Is "second order intermittency?". Is this a formal concept?

What we mean here is that during the intermittent phases, the PM dynamics oscillate in the range (0.2 1) with an average of about 0.6. In the non-intermittent phases, the PM dynamics is stuck near 0. Therefore, the intermittency is on average (shift from 0.6 to 0) and in variance. This explanation has been added to the new version of the manuscript.

L270. Forward Euler time steppers are notoriously inaccurate. Why not use a more advanced time stepper (e.g. Runge Kutta) for better accuracy? There are many convenient software packages for integrating ODEs with better schemes (e.g. ode45 in MatLab). What is N? It must be network size, but given all the notational changes it is hard to be sure.

We remind that the idea is here to have exemples close to the atmospheric or climate data: when considering daily or 6 hourly data, as commonly done in climate sciences and analyses, we hardly are in the case of a smooth RK time stepper. We therefore stick to the Euler method for similarity with the actual climate data. This has been added to the text.

L331: "We show the results using the residuals (Eq. 9)" Why not show the results with the "exact method" (Eq. 11)? It seems the earlier results implied this technique was more effective.

Unfortunately the "exact method" cannot be used for the SLP NCEP data. Indeed this dynamics has a spatial component that the "exact" method cannot take into account the spatial component. This is now specified in the paper

Figure 10 b-d. These panels all look different. I don't see much reason to prefer panel d to c. Could the authors present a more convincing visualization for the claimed improvement of the moving average filter? Maybe a single power-spectra plot would be more succinct, especially since the author's don't comment on the timing of the high-frequency vs low-frequency results.

The wavelet methodology is a more sophisticated representation of a spectrum. We have replaced wavelet spectra with conventional spectra (now Figure 10b). For consistency, figure 10a is also replaced with the pdf for the observable u, used in figure 9 and in figure 10b. This means that all diagnostic on SLP are now computed using u(t)=<SLP(t)>\_{lon,lat}, i.e. the time series of the spatial average of SLP

L373. "For the Lorenz 1996 mode, we did not apply a moving average filter to the data,..." It would have been nice to see this motivation described in Section 3.

**We have added this motivation in Section 3**

**TECHNICAL CORRECTIONS**

L73. 'Reservoir compution'' There is a missing quote.

L74. "The principle of Reservoir computing" Does "Reservoir" need to be capitalized here? If so, I would expect "computing" to be capitalized as well. "reservoir" is not always capitalized in this manuscript. L76. "In our study ESNs are implemented"

L77. "The code is given in the appendix

L97: "to this purpose" -> "for this purpose"

L239: "we find the best match. . . are obtained for w=3" Correct "are" to "is". 249: "Figure 5a)" Remove the parenthesis

Line275. "Figure 6.b,d)" This should read "Figure 6 b,d".

Figures should be referred to with a consistent convention.

L288. "distance T". C6 Do the authors mean  $\Sigma$ ? T is the length of the time series. Figure 8: The text in this graphic is fuzzy. Please save at a higher resolution.

Figure 2a: This plot has too many curves. Red-green is bad for colorblind readers. It is hard to see the author's point.

Figure 3, 4: These colorscales are not legible for colorblind readers. I could not interpret these figures and relied on the author's textual description of the results. I suggesting using "viridis" or another sequential colorbar.

**Thank you, technical corrections have been implemented. Colorscales replaced as demanded for colorblind readers.**

We thank the reviewer for the references, which have been properly added to the new version of the manuscript.

**RC2 - Anonymous Reviewer**

The authors are utilizing Echo State Networks to predict filtered dynamics in the perturbed Lorenz 1963 equations, the Pomeau-Manneville 89 intermittent map, and the Lorenz 1996 equations. A moving average filter is utilized for scale separation in time. The filtered dynamics are smoother and easier to predict. A residual term is added, either sampled from the training data, or based on an analytic formula derived from the moving average filter. Assuming that the filter width is smaller than the associated large timescales of the processes involved, the large scale processes can be successfully predicted. The authors claim that modeling only the spatially coarse grained and time averaged state can boost performance of ESN. However, the generalization of this argument to more realistic systems is not sufficiently supported by the results, as elaborated in the comment section below. The idea of utilizing a moving average filter for noise reduction and scale separation, or spatial coarse graining is known. I am not sure that the novelty of the paper to apply ESNs to (spatially/time) filtered dynamics, is enough to guarantee publication in the journal. The effect of the unmodeled dynamics (the information lost during filtering) is not taken into account in the model. In most interesting applications, the effect of the unmodeled modes is the problem, and a field of study by itself (closure models in turbulence, small scale models in weather etc.)

We respect the opinion of the reviewer on our work, but we feel that the motivations provided here and in the following comments for rejection are not supported and sometimes do not contain any element that could help us to improve the manuscript. For example, the reviewer states many times that our results are "known" or "existing in the literature" or "hardly surprising" but not a single reference to previous works which should contain our results is provided. We are therefore unable to assess which part of our work could be not original or to provide an adequate rebuttal to state why our work is instead original. Furthermore, the reviewer says that our results are "not sufficiently supported by the results", but it is not said in which way this is the case. We stress that, for all cases presented, we have used at least three statistical metrics to assess performances, and scanned a large range of coarse graining and noise intensities, as well as performed several realisations of our systems. If this is not enough to warrant publication, then we would like to know why our suggested metrics are not sufficient to support our conclusion. Besides these remarks on the structure of the proposed comments, we will do our best to answer the reviewer's comments and thus to improve the manuscript.

1 Comments

1. In the three-dimensional Lorenz system, it is logical that the moving average filter produces better results. By construction, noise is added to the system. It does not come as a surprise that the ESN predicting the filtered dynamics (which are smoother) and augmented with the random residual terms, shows superior performance. However, there is no complex multiscale effect taking place, as the whole state information is given to the system (no hidden state, at least nothing is mentioned in the text about it). Moreover, as a reference time-scale, the Lyapunov time of the deterministic system is used, although the system is augmented with noise, which means that the effective Lyapunov time is in essence much shorter, as stochasticity accelerates the divergence of nearby trajectories. In any case, it is important to be critical about the conclusions drawn from this case.

We agree with the reviewer that "it is logical that the moving average filter produces better results" but what we would like to show is that there is a dependence on the noise intensity on the quality of the results obtained and particularly that the moving average filters is very useful for intermediate noise intensity, namely when the stochastic component starts to affect the deterministic dynamics, but not at the same order of magnitude. We disagree with the statement that " the effective Lyapunov time is in essence much shorter, as stochasticity accelerates the divergence " as this is also dependent on the noise level. The most interesting performances for the filtered ESN are obtained when the noise is yet 2-3 order of magnitude smaller than the typical scales of the deterministic component, and we expect the perturbation on the lyapunov exponents to be of these orders as well. In the conclusion section we have added: Most previous studies on ESNs were handling relatively smooth signals, and not such rapidly changing signals. Although it does not come as a surprise that utilizing the ESN on the time averaged dynamics and then adding a stochastic residual improves performance, the main insights is the intricate dependence of the ESN performance on the noise structure and the fact that, even for non-smooth signal, ESN with hyperbolic tanh functions can be used to study systems that have a multiscale dynamics. In the new version of the manuscript we have added these considerations to the discussion section of the articles.

2. In the Pomeau-Manneville intermittent map, it is not a surprise that the ESN cannot capture the dynamics, as they are changing very rapidly, even visually they look completely stochastic. A deterministic ESN with tanh (smooth, continuous) activation function cannot be expected to produce trajectories that look spiking/stochastic/rapidly changing. Most previous studies on ESNs were handling relatively smooth signals, and not such rapidly changing signals. At least the nature of the signal has to be taken into account in the selection of the activation function of the reservoir. Thus, it does not come as a surprise that utilizing the ESN on the time averaged dynamics and then adding a stochastic residual improves performance. As expected, the plain ESN diverges, as demonstrated also in previous studies with such non-smooth signals.

We thank the reviewer for the comment. Indeed this can be a good explanation of our results. The reviewer says again that "it is not a surprise" or "demonstrated also in previous studies". However, no references are provided for us to improve the quality of the manuscript or to give credits to those who have already analysed this problem on another angle. We would be more than happy to include and discuss those references in the manuscript. Furthermore, the reviewer makes a confusion between a visual analysis and what signals truly are. The PM system is piecewise continuous & differentiable, and is hence "relatively" smooth from the mathematical point of view.

3. In the Lorenz 96 system, as demonstrated in Figure 8, the method fails to capture the long-term climate, as the dynamics predicted by the ESN are clearly different from the groundtruth.

We only partially agree with the reviewer about the results obtained for Lorenz96. Although the detailed dynamics does look different from that of the original system, there are a few things correctly captured by the ESN, namely the quasiperiodic spatio-temporal oscillations and the fact that ESN produces non-divergent dynamics.

4. In the sea-level pressure, the moving average filter ESN does not achieve any significant improvement based on the results in Figure 9.

Here, we would like to gently disagree with the referee comment. The ESN with filter does produce significant improvements, in terms of all the metrics considered, and as noted by the other reviewer.

5. In the abstract, the authors claim that "multiscale dynamics and intermittency introduce severe limitations on the applicability of recurrent neural networks, both for short-term forecasts, as well as for the reconstruction of the underlying attractor". This is shown for Echo State Networks in the document, but not in general for Recurrent Neural Networks. The argument has to be relaxed to take into account only ESNs, or a relevant reference for other RNN architectures should be given.

**We agree with this comment of the referee, in the new version of the manuscript we have clearly restricted our attention to ESNs.**

6. There is a contradiction in the text, in page 3, the authors state that "We aim at understanding this sensitivity in a deeper way, while assessing the possibility to reduce its impact on prediction through simple noise reduction methods", although one sentence before, they claim that they choose the ESN framework for "...its ability to forecast chaotic time series and its stability to noise". These sentences are contradicting each other. Later in the text, the authors state "Since Echo State Networks are known to be sensitive to noise (see e.g. [34]), ...".

What we mean here is that ESN are less sensitive to noise than other techniques, but also that our goal is precisely to evaluate such sensitivity and the improvement coming from noise reduction techniques.

7. The analysis of the performance of the proposed method based on different parameters e.g. intermittency of dynamics/degree of coarse graining, etc. is interesting. However, this is not adequate to warrant publication.

We are delighted to see that the reviewer admits that our results are interesting. This encourages us to pursue their publication.

**Boosting performance in Enhancing geophysical flow** machine learning of geophysical flows performance via scale separation**

Davide Faranda1,2,3, Mathieu Vrac1, Pascal Yiou1, Flavio Maria Emanuele Pons1, Adnane Hamid1, Giulia Carella1, Cedric Ngoungue Langue1, Soulivanh Thao1, and Valerie Gautard4

[revised manuscript text omitted]

95
$$\underline{r}\mathbf{r}(t+dt) = \tanh(\underline{W}\underline{r}\underline{W}\underline{r}(t) + W_{in}\underline{u}\underline{x}(t)),$$
 (1)

where W is the adjacency matrix of the reservoir: its dimensions are  $N \times N$ , and N is the number of neurons of in the reservoir. In ESN, the neuron layers of classic deep neural networks are replaced by a single layer consisting of a sparsely connected random network, with coefficients uniformly distributed in [-0.5; 0.5].  $W_{in}$ , with dimensions The  $N \times K_{\tau}$ -dimensional matrix  $W_{in}$  is the weight matrix of the connections between the input layer and the reservoir, and the coefficients are randomly sampled, as for W. The output of the network at time step t + dt is

$$W_{out}\underline{r}\boldsymbol{r}(t+dt) = \underline{v}\boldsymbol{y}(t+dt)$$
(2)

where  $\frac{v(t+dt)}{y(t+dt)}$  is the ESN prediction,  $W_{out}$  with dimensions  $K \times N$ , is the weight matrix of the connections between the reservoir neurons and the output layer. We estimate  $W_{out}$  via a ridge regression (Hastie et al., 2015):

$$W_{out} = \underline{v} \boldsymbol{y} (t + \underline{dt} \leq \underline{T}) \underline{r} (t + \underline{dt} \leq \underline{T})^T [\underline{r} (t + \underline{dt} \leq \underline{T}) \underline{r} (t + \underline{dt} \leq \underline{T})^T - \lambda I]^{-1}$$
(3)

105 with  $\lambda = 10^{-8}$ . Note that we have investigated different values of  $\lambda$  spanning  $10^{-8} < \lambda < 10^{-2}$  and found no sensitive differences in the performance of ESN. In the prediction phase we have a recurrent relationship:

$$\underline{\mathbf{r}}\mathbf{r}(t+dt) = \tanh(\underline{W}\underline{\mathbf{r}}\underline{W}\underline{\mathbf{r}}(t) + W_{in}W_{out}\underline{\mathbf{r}}\mathbf{r}(t)).$$
(4)

[revised manuscript text omitted]

(8)

We can define the residuals as:

175
$$\delta \underline{u} x(t) = \underline{u} x^{(f)}(t) - \underline{u} - x(t).$$
(9)

In practice, the computation always refers to the discrete time case, as continuous time systems are also sampled at finite time steps. Since Echo State Networks are known to be sensitive to noise (see e.g. Shi and Han (2007))(see e.g. Shi and Han, 2007), we exploit the simple moving average filter to smooth out high-frequency noise and assess the results for different smoothing windows w. We find that the choice of the moving averaging window w must respect two conditions: it should be large enough to amount the noise but arealise than the choice of the generativistic time  $\pi$  of the large case fluctuations of the custom. For chapter

180 to smooth out the noise but smaller than the characteristic time  $\tau$  of the large-scale fluctuations of the system. For chaotic systems,  $\tau$  can be derived knowing the rate of exponential divergence of the trajectories, a quantity linked to the Lyapunov exponents (Wolf et al., 1985), and  $\tau$  is known as the Lyapunov time.

We also remark that we can express explicitly the original variable u(t) variables x(t) as a function of the filtered variable 185  $u^{(f)}(t)$  variables  $x^{(f)}(t)$  as:

$$\underline{u}\boldsymbol{x}(t) = w(\underline{u}\boldsymbol{x}^{(f)}(t) - \underline{u} - \boldsymbol{x}^{(f)}(t-1)) + \underline{u}\boldsymbol{x}(t-w).$$
(10)

we We will test this formula for stochastically perturbed systems to evaluate the error introduced by the use of residuals  $\delta u \delta x$ .

**2.3 Testing ESN on filtered dynamics**

- 190 Here we describe the algorithm used to test ESN performance on filtered dynamics:
  - 1. Simulate the reference trajectory u(t) x(t) using the equations of the dynamical systems, where u(t) has been standardized and standardize x(t) by subtracting the mean and dividing by its standard deviation.
  - 2. Perform the moving average filter to obtain  $\frac{u^{(f)}(t)x^{(f)}(t)}{u^{(f)}(t)}$ .
  - 3. Extract from  $\frac{u^{(f)}(t)}{x^{(f)}(t)}$  a training set  $\frac{u^{(f)}_{train}(t)}{x^{(f)}_{train}(t)}$  with  $t \in \{1, 2, \dots, T_{train}\}$ .
- 195 4. Train the ESN on  $\frac{u_{train}^{(f)}(t)}{u_{train}^{(f)}(t)} \frac{x_{train}^{(f)}(t)}{u_{
[revised manuscript text omitted]

15

---

## Author Response (AR2)

**Editor Decision: Publish subject to minor revisions (review by editor)** (21 Jun 2021) by Amit Apte

Comments to the Author:
Dear authors, As you will see from the reviews, there are several changes suggested by the reviewers that I feel will be quite helpful for improving the manuscript. In particular, it would be great if you can address the following comments from report #2 that I quote: "Improved notation; Simpler plots to accompany the detailed ones; A more careful study of the effects of regularization." Based on the available reviews, I am suggesting publication subject to review by the editor.

**Dear Editor,**

**Thank you very much for your help with this paper and no worries for the delay in the reviewing process. We provide below a complete answer to both reviewers' queries. We do hope that the current version of the manuscript, improved as detailed in the markedup attachment, will be suitable for publication in NPG. We have taken extreme care in fixing the notation as requested by the reviewer. However, we do feel that the dependence of the results on the network size is an interesting outcome of our study, especially in climate sciences where there is a clear alternative to Machine Learning methodologies, namely using the underlying equations. As explained in the answers below, we prefer not to add regularization to explicitly show the cases where overfitting is produced.**

**Best wishes,**
**Davide Faranda**

**REPORT #1**

The authors argue that there are serious limitations for applicability of out-of-the-box Echo-state network (ESN) in geophysical flows characterized by intermittency and multiple temporal scales, both for short-term forecasts and long-term attractor prediction. The performance can be improved by training ESN on the time averaged dynamics of large scales, and adding stochastic component accounting for small scales. The results are presented for 3 toy models and sea-level pressure dataset, and are convincing overall. I generally like the paper and it can be published once the following issues are addressed.

**We thank the reviewer for the positive feedback on our work. We have taken into account the following comments.**

1. Deep learning methods with add-on stochastic component have been explored in references below, those should be cited as part of the review in introduction.

Mukhin, D., Kondrashov, D., Loskutov, E., Gavrilov, A., Feigin, A., & Ghil, M. (2015). Predicting critical transitions in ENSO models. Part II: Spatially dependent models. Journal of Climate, 28(5), 1962–1976.

Seleznev, A., Mukhin, D., Gavrilov, A., Loskutov, E., & Feigin, A. (2019). Bayesian framework for simulation of dynamical systems from multidimensional data us- ing recurrent neural network. Chaos, 29(12), 123115. doi: 10.1063/1.5128372

**The references have been added to the manuscript.**

2. The raw FFT power spectra in Fig.5&10 are too noisy and overlap, and thus very difficult to compare and interpret. Suggest to use spectral methods with proper smoothing, for example Multitaper Method.

**We thank the reviewer for the suggestion. MTM methods would indeed be interesting for quantitative estimates of spectral features. Here, we stay rather qualitative in the spectral description. We therefore prefer to keep the present presentation for those figures.**

**REPORT #2**

Overall, this paper illustrates important points about inadequacies in an existing data-driven approach (reservoir computing) to modeling complex chaotic systems with intermittencies and couplings to unobserved (multi-scale) processes. I agree that this work ought to be published and have the following recommendations to make the work more clear and impactful:

1. Improved notation

2. Simpler plots to accompany the detailed ones

3. A more careful study of the effects of regularization

**Dear Matthew, thank you for your review of the paper and your suggestions. We have taken care of answering your comments below.**

**Detailed comments**

Comment on ESNs and memory:
I suggest you distinguish between ESNs and other RNN approaches.
See work by Pantelis Vlachas https://arxiv.org/abs/1910.05266
which suggests that although ESNs have the capability for memory, they often struggle to represent it when compared to fully-trained RNNs.
Note that your references to Shi and Han 2007 and Li et al 2012 all use ESNs with delay-embedding representations.
This essentially defeats the purpose of ESNs, as they are supposed to "learn" this.
A caveat in those papers is the Mackey-Glass DDE, but this is a very simplistic type of memory that can easily be stored by an ESN.
From the work by Vlachas, others, and my own experiments, I have deep suspicions about whether ESNs can learn memory meaningfully at all.
So, I do wish that all the reported experiments could be repeated with a simple ANN, GP regression, Random Feature Map, or other data-driven function approximator that does NOT have the dynamical structure of RNNs/ESNs.
These other approaches I list are generally much easier to train and tune than ESNs and have a more clear interpretation.
In particular, the authors might be interested in Random Feature Map approaches (see Gottwald and Reich 2020 treatment of RF-based regression https://arxiv.org/abs/2007.07383),
as it is essentially an ESN without the recurrence (and is a universal approximator for C1(Rn,Rn) ).
I am not requesting an entirely new study with new methods, but a discussion of why ESNs are chosen ought to consider the perspective I outlined above.

**Thank you, we have added and discussed this issue in the conclusion section: "Our results, obtained on ESN, should also be distinguished from those obtained using other RNN approaches. Vlachas et al.(2020); Albers et al. (2018) suggest that, although ESNs**

have the capability for memory, they often struggle to represent it when compared to fully-trained RNNs. This essentially defeats the purpose of ESNs, as they are supposed to learn memory. In particular, it will be interesting to test whether all experiments reported here could be repeated with a simple artificial neural network, Gaussian Processes regression, Random Feature Map, or other data-driven function approximator that does not have the dynamical structure of RNNs/ESNs (Cheng et al., 2008; Gottwald and Reich, 2021

Equations for ESN:
the "t<T" notation is somewhat unclear...do these indicate a matrix? Perhaps capital letters would be better?

**We have revised the notation that was indeed unclear.**

Notes on regularization (line 96)
In what settings were the values of lambda investigated?
Why limit to 10^-8? Perhaps 10^-9 is better?
More importantly---I am quite surprised that there was no effect of lambda on performance, given that many results show a "sweet spot" for the network size...larger networks performing worse is a classic sign of overfitting that can be addressed with increased regularization.

**We have rephrased this sentence as: " Note that we have investigated different values of $\lambda$ spanning $10^{-8} <\lambda < 10^{-2}$ on the Lorenz 1963 example and found little improvement only when the network size was large, with $\lambda$ partially preventing overfitting. Values of $\lambda<10^{-8}$ have not been investigated because too close or below the numerical precision"**

Statistical distributional test:
I found this setup rather difficult to read, and the ultimate choice to use Monte Carlo approximations unclear and potentially statistically unsound.

**See answer below about Monte Carlo.**

-To begin, it would help to define zeta as a function mapping between two spaces. Where does u live? Where does x live? How does this relate to y?

**We have added to the text that zeta is a function mapping between two spaces. However, we would like to stress that the choices of x (the dynamical system) and u (the observable) are rather arbitrary and they could live in any metric space allowed in dynamical systems theory. The relation between x and y is obtained by substituting r(t) in Eq. 2 with the expression of Eq. 1.**

What is x(t)? This is the first time it appears in the paper (line 108).

**In the previous version of the paper, x(t) was already defined in line 85 as "Let $\vec{x}(t)$ be the $K$-dimensional observable consisting of $t=1,2\dots,T$ time iterations, originating from a dynamical system, and $\vec{r}(t)$ be the $N$-dimensional reservoir state"**

Is R_U in R^1? R^N? a banach space?

**R_U and R_V are Banach spaces, whose dimension depends on the choice of the observables. This has been added to the text.**

What is meant by the "marginal distribution of the forecast sample"? Marginal over what?

**We use the term "marginal distribution" in the time series analysis/stochastic process sense, i.e. the marginal distribution is the probability density function of the observed sample taken as i.i.d., as opposed to the joint distribution, which includes the covariance structure of the stochastic process. For example, an autoregressive process with innovations N(0,sigma) will have a marginal N(0, sigma^2/(1-phi^2)) distribution, but a joint multivariate Normal distribution with geometrically decaying autocorrelation.**

-The Monte Carlo approach also confused me---please clarify this a bit more.
Where is the randomness in samples coming from? in time? across initial conditions?
Also what is the goal of the MC process? Is it to estimate a baseline \Sigma for f vs \hat{f}?

**The rationale behind adopting a Monte Carlo method follows the need for statistical testing the distributional equivalence while dealing with two main problems, that are stated at lines 125-135: we cannot always use the full histogram because of empty bins, and we cannot observe the invariant distribution of the simulated systems, even with very long simulations. These two shortcomings are expected to produce deviation from the assumptions that make the test statistic actually follow a chi-squared distribution.**

**It is very common in modern statistical inference to use bootstrap resampling to better approximate the asymptotic distribution of an estimator or of a test statistic in presence of small or noisy datasets (see, e.g. the book by Davison & Hinkley, Bootstrap Methods and their Application 1997, Cambridge University Press). In our case, since we are dealing with simulated systems, we do not need to rely on resampling, and we can instead produce large ensembles (here 10^5 trajectories) and tabulate the observed distribution of the test statistic under the null hypothesis to obtain critical values.**

**Concerning the randomness, there are two sources, as explained at lines 228-230: the system is perturbed with an additional noise, and each trajectory starts from randomly sampled initial conditions. This guarantees a source of randomness even in cases where the amplitude of the noise is 0. This is true for Lorenz 1963 and the Pommeau-Manneville map, while the Lorenz 1996 always has a perturbation term in the first mode (lines 305-310).**

-Throughout the paper, neither Sigma nor phi take on a physical meaning for me---it seems that they are simply used to compare methods based on their ability to reproduce statistical quantities. The statistical validity of chi-squared does not seem to be important in the paper.

**Indeed Sigma has no physical meaning and it is defined only for statistical purposes. However, it does have a statistical validity, as it is used to perform an inferential test. Even though we use tabulated critical values under H0 instead of the theoretical chi-squared quantiles, this is only done to overcome limitations in the assumptions. Also phi does not have a physical meaning, as it is rather a purely statistical performance measure for the ESN.**

Due to this, I would highly recommend using Kullback Leibler-divergence instead of the chi-squared metric---it measures the information loss between probability measures and seems more suited to the job in the paper.
However, I understand this may be significant extra work---perhaps just make a comment that KL could be another option?
Still, I think this section would be much clearer if it were built around KL-div.

**The initial choice of the chi-squared was based on the need to perform a statistical test; while the KL divergence is indeed the most theoretically founded divergence measure**

between probability distributions, its sample version is not a statistic with a defined probability distribution, so that it doesn't allow to obtain confidence intervals.

Both the KL and the chi-squared divergences are non symmetric, an ideal property when measuring a "proximity" to a reference probability distribution (see e.g. Basu et al., 2019 about the use of non-symmetric divergences in statistics).

The KL divergence is rarely used to directly define estimators or test statistics, even though it is implicitly used, since in the maximum likelihood (ML) approach, the parameters estimated with ML produce the statistical model (in the selected class) with the minimum KL divergence from the one generating the sample. Other than in estimation, KL is also linked to hypothesis testing, as it can quantify the loss of power of likelihood ratio tests in case of model misspecification (Eguchi et al., 2006).

It has been known for a long time that statistical estimation based on minimum chi-squared is equivalent to other popular objective functions, including the log-likelihood and, thus, the minimum KL (see Berkson, 1980 ).

Our choice was, indeed, based on the statistical validity of the chi-squared as a meaningful metric that allows performing frequentist statistical testing. In particular, one advantage of the chi-squared is that it is independent of the pdf of the distributions to be compared.

We added this explanation to the text, even though we find it could add confusion rather than clarity, since the chi-squared is a very standard goodness-of-fit test statistic, while the KL is very rarely used in such a setting.

Basu, A., Shioya, H., & Park, C. (2019). *Statistical inference: the minimum distance approach*. Chapman and Hall/CRC.

Berkson, J. (1980). Minimum chi-square, not maximum likelihood!. *The Annals of Statistics*, *8*(3), 457-487.

Eguchi, S., & Copas, J. (2006). Interpreting kullback–leibler divergence with the neyman–pearson lemma. *Journal of Multivariate Analysis*, *97*(9), 2034-2040.

-Also, in this section you should give the examples of zeta that are used---1st coordinate and sum of all coordinates. This will help the reader anticipate what is to come.

**we have added this to the text**

APE:
-The formulation of s is interesting, as it can equivalently be written as \Delta t times the time average of the derivative \dot{u}.
Would it make sense to then have more stringent demands on divergence when the timestep shrinks?
In the dt -> 0 limit, s->0 which seems odd.
Please justify this choice with respect to the chosen sample rate.

**Thank you. We have added to the text the alternative definition of APE. Here we intend the divergence only for fixed timestep, so it is to be intended as a statistical metric rather than an asymptotic quantity.**

-Also tau_s is not defined (line 148) (or I could not find it).

**The definition was given just below in a sentence, but we have now written a specific equation (7) for clarity**

Moving average filter:
-perhaps note that the the average can be left/right or centered, but I suppose you choose right side of the window since you want to keep the system Markovian.

**This is right, in the manuscript this reads: "The moving average operation is the integral of $u(t)$ between $t$ and $t-w$, where $w$ is the window size of the moving average."**

2.3 Testing ESN
When reading the algorithm 1-8, I became confused with y(t) vs x(t). Please make explicit what is data / truth and what is coming out of the ESN---is the ESN trained on and predicting the full state? Or an observable?

**thank you for your remark. We have added: "Note that the relation between y(t)and x(t) is given in Eqs 1-2."**

Step 5---is the ESN forecast always produced directly after a training trajectory?
This seems like a rather limiting case---it would be better to re-initialize the trained ESN on a new trajectory...how do we know each ESN hasn't overfit to a small region of state space?

**Yes we have always produced ESN directly after a training trajectory. We have tested starting the trajectory elsewhere for the Lorenz 1963 attractor and we found that the ESN forecasts depend in a non-trivial way from the chosen point. We would like to explore this interesting dependence in a further study.**

Step 7: Why does u depend on the future t>Train? This t>T notation is a bit confusing to me, and needs to be defined explicitly.

**We have now corrected the notation. We have added "Note that $\vec{x}(T_{train}<t<T)$ is the ensemble of ~\texit{true values} of the original dynamical system."**

Step 11: Please define v(f) explicitly. Also, why does this equation hold? I see how (10) is true because xf is defined in terms of x. But in (11) vf is defined by yf (I assume) which is an output from ESN. Is (11) only approximate? Please clarify.

**We have added v(f) definition in Step "11". It is true that equation 10 is always valid but Eq 11 is only approximate because of the filtering procedure. This is now specified in the text.**

L63
210: are they iid? **Added to the text**
220: green -> black **Corrected**
Fig 2: Where do the many trajectories come from? Different trained ESNs (on the same or different training sets)? **same training set, this has been specified**
Fig 3b: For large noise, why do smaller networks work better? Is this an overfitting problem that can be fixed with regularization? Same question for 3c...if not overfitting, perhaps larger networks need longer time to initialize?

**\*From this and subsequent comments, we understand that the reviewer does not consider as a main outcome of our work the dependence of the results on *N*, mostly because we could add regularization to fix this. While we understand and respect the reviewer's viewpoint, we would like to stress that the dependence on *N* can be of great interest for climate applications, as we also look for the most parsimonious ML model to perform certain tasks. Indeed, with respect to other scientific fields where ML techniques are the only way to explore the behavior of complex systems, for climate sciences we know the evolution equations of the dynamics, so that ML techniques should also be appealing from a computational point of view to act as a substitute for equations. Furthermore, adding regularization may solve the overfitting issue , it won't clearly improve the results - at least for the examples considered here. We would therefore keep the figures and the results displayed in terms of both N and noise intensity for the exemples considered.**

Fig 3 overall: It is hard to concretely connect phi (and sigma) to the quality of the estimated invariant statistic.
It would help to plot a blue/green/yellow KDE vs the true KDE so that we can see what phi is really discriminating between.

**These two indicators are designed to give a statistical measure of the large-ensemble properties of ESN predictions, rather than to quantify the quality of the single trajectory. As already mentioned, \Sigma is a traditional test statistics to evaluate the proximity of histograms, and \phi is simply defined as the large-ensemble average rejection rate of the test, which gives a failure rate of the ESN in reproducing statistically equivalent distributions.**

**Concerning the distribution plots, direct comparison of the empirical pdfs or cdfs may indeed give an idea of what type of deviations happen in a single prediction. However, this is a simulation study based on 10^5 trajectories of three different variables, so a visualization as described, and as shown in Fig. 10a for the SLP data, does not appear to be feasible.**

Finally, I feel these plots could be much simpler by collapsing over (or fixing) N---this is only possible if you can pick a large enough N and that regularization can prevent the "sweet spot" issues wrt N.

**We thank the reviewer for this point, however it is our precise choice to show the behaviour of the ESN as a function of the network size, to investigate its stability and flexibility and for the reasons explicitly given in comment (\*)**

PM map
Great idea to study this system. L63 is very common and often "too easy".
246: This is not a deterministic map, correct? Also, notation \xi_t would be more consistent than \xi(t).

**The original PM map is deterministic, our addition makes it stochastic. We keep the notation \xi(t) for consistency with the Lorenz 1963 example where the notation \xi_t would imply a double subscript.**

Fig 4: Why is tau_s better for smaller networks?

**It appears that in smaller networks, the ESN better tracks the initial conditions, so that the ensemble shows smaller divergences. This has been added to the text.**

Again, I hypothesize this is an issue of regularization. Also, again, this figure feels like information overload for me---if possible, this could be easier to understand when fixed/collapsed over N (and this becomes a supplementary figure).

**Thank you, however again we would opt for keeping visible the dependence on *N* for the reasons explicitly given in comment (*)**

Comment: Is the intermittency you observe driven more by the noise or the deterministic PM map itself? I wonder if the RC performs poorly due to the randomness of the intermittency (driven by \xi_t), rather than the intermittency itself.

**It is driven by the deterministic map. We have added "deterministic" in the text. So that the intermittency is not random but driven by the chaotic behavior.**

Can ESN handle the PM map with epsilon=0? Why or why not? This seems like a very simple problem that already breaks the data-driven method. Further comment on what is wrong or what needs to be studied would be very valuable here. Also is log the natural or common log here?

**When in the text we discuss the deterministic limit, we mean epsilon=0, this means that ESN cannot handle the deterministic PM map. We can therefore speculate that there is an intrinsic problem in reproducing intermittency driven by the deterministic chaos. Log is the natural logarithm, thus it has been added to the text.**

L96
In this problem, we finally have unobserved scales (and, hence, memory)---this brings up the challenge of memory I mention earlier and the work of Vlachas et al.
I would be very curious whether you'd see different results with a vanilla ANN; alternatively, I wonder if LSTM (which seems to have more hope of retaining memory) would perform differently. Comments on this would be welcome.

**Thank you for the appreciation of these results. We have indicated the ANN analysis as a possible follow-up study of this paper.**

304-305: Please clarify this statement.

**We have rephrased as: "The rate of failure $\phi$ is very high (not shown) because even when the dynamics is well captured by the ESN in terms of characteristics time and spatial scales, the predicted variables are not scaled and centered as those of the original systems"**

Fig 6: Excellent Figure!!! **Thank you**
Fig 7: My main complaints arise again here:
1) The dependence on N feels secondary to the main point of this plot, which compares performance for different c, h and X vs X,Y.
So, again, perhaps fix an N and show a box plot?
2) 1 can only be done safely if the N-dependence is simpler...which, again, I hypothesize can be addressed via regularization.
If not, please show this. Currently, the results are confusing, as they show many performance metrics worsening for larger networks---this is in-line with an overfitting hypothesis.
3) Showing the actual KDEs will bring much more light to the differences in Sigma.
Also, are APE and Sigma normalized w.r.t. dimensionality? Their definitions do not suggest so, and thus I wonder if X vs X,Y results are fair comparisons. Please double check and clarify in the text.

**We thank the reviewer for these points, however it is our precise choice to show the behaviour of the ESN as a function of the network size, to investigate its stability and flexibility, showing overfitting. The reasons explicitly given in comment (\*) apply.**

Fig 8: This figure is illuminating, as it shows the explicit dynamcs. The evaluation metrics are excellent for high-throughput comparisons, but their meanings can get lost.
How should we interpret Fig 8? Is this better/worse than other methods? Is this a pathologically good/bad example? Or is it average? For contrast, Vlachas et al. seems to have much more realistic looking ESN fits.

**We have added:"This figure shows an average example of the performance of ESN in reproducing Lorenz 1996 system when the fit succeeds. For comparison, we refer to the results by Vlachas et al (2020) which shows that better fits of the Lorenz 1996 dynamics can be obtained using back-propagation algorithms."**

NCEP
Fig 9: How can Sigma blow up wrt N but tau_s and eta stay stable?
Is there an issue with long-time stability of the ESNs that is not shown? Vlachas et al. reported on this issue.

**Yes, indeed the blow-up is related to the long-time stability of the ESN. The blow only affects global indicator Sigma and not tau_s and eta which refers to short term properties. This has been added to the text.**

**other**
Line 73: should be "Finally, we"
Line 284: "the the"

**Thanks, corrected**

--Matthew LevineR